# Rbpms2 promotes female fate upstream of the nutrient sensing Gator2 complex component Mios

Miranda L. Wilson[1], Shannon N. Romano[1], Nitya Khatri[1], Devora Aharon[1], Yulong Liu[2], Odelya H. Kaufman[3], Bruce W. Draper [2] & Florence L. Marlow [1,3] ✉

Reproductive success relies on proper establishment and maintenance of biological sex. In many animals, including mammals, the primary gonad is initially ovary biased. We previously showed the RNA binding protein (RNAbp), Rbpms2, is required for ovary fate in zebrafish. Here, we identified Rbpms2 targets in oocytes *(Rbpms2-bound oocyte RNAs; rboRNAs)*. We identify Rbpms2 as a translational regulator of *rboRNAs*, which include testis factors and ribosome biogenesis factors. Further, genetic analyses indicate that Rbpms2 promotes nucleolar amplification via the mTorc1 signaling pathway, specifically through the mTorc1-activating Gap activity towards Rags 2 (Gator2) component, Missing oocyte (Mios). Cumulatively, our findings indicate that early gonocytes are in a dual poised, bipotential state in which Rbpms2 acts as a binary fate-switch. Specifically, Rbpms2 represses testis factors and promotes oocyte factors to promote oocyte progression through an essential Gator2-mediated checkpoint, thereby integrating regulation of sexual differentiation factors and nutritional availability pathways in zebrafish oogenesis.

Oogenesis is an essential process for all sexually reproducing organisms. Successful oogenesis is complex and demanding, requiring immense cellular growth, high transcriptional and translational burdens, and the generation of a large reserve of maternal factors necessary for the embryo. In some organisms, such as humans, oocytes are very long-lived cells that must tightly regulate their development and maturation in response to internal and external cues (reviewed in ref. 1). Our lab has previously shown that the RNA binding protein of multiple splice forms 2 (Rbpms2) is required for oocyte development and subsequent female sex differentiation in zebrafish but its mode of action and targets remain unclear[2,3]. Based on our prior findings that Rbpms2 protein is expressed in oocytes but not detected in somatic gonad or testis[2], is required for ovary development, and promotes ovary fates independent of the checkpoint regulators Tp53 and Chk2[2], we envisioned two non-mutually exclusive hypotheses for

Rbpms2 functions in ovary specific differentiation of germ cells. (1) Rbpms2 actively represses translation of testis factors that are expressed in the early germline cells of ovaries and (2) promotes translation of factors that allow for oocyte growth and differentiation upstream of an unknown oocyte progression checkpoint.

Early oogenesis relies on a high degree of transcription and generation of translation machinery components, like ribosomes, to allow for extreme growth and support the translational needs of oocytes (reviewed in ref. 4). Therefore, adequate nutrition is necessary for oogenesis (reviewed in ref. 5). The interplay of oocyte development, nutrition availability, and fertility has been observed across species[6–8], including zebrafish where a low nutrition environment biases sexual development to male[9]. In other species, low-nutrition environments can suppress or arrest oogenesis to preserve energy for homeostatic mechanisms[6–8].

[1]Department of Cell, Developmental, and Regenerative Biology. Icahn School of Medicine at Mount Sinai. One Gustave L. Levy Place Box 1020, New York, NY, USA. [2]Department of Molecular and Cellular Biology. University of California. 1 Shields Ave, Davis, CA, USA. [3]Department of Developmental and Molecular Biology. Albert Einstein College of Medicine, Bronx, NY, USA. ✉e-mail: florence.marlow@mssm.edu

Nutrition sensing pathways are important to the viability of all cells. For example, high levels of fatty acids and carbohydrates are necessary for the activation of the citric acid cycle for adenosine triphosphate (ATP) production, whereas glycolysis relies on sufficient glucose intake. Other mechanisms respond to sufficient amino acid levels, like the metabolic homeostasis regulator mechanistic target of rapamycin (mTOR) pathway (reviewed in ref. 10). Through amino acid detection, the Gap activity towards Rags 2 (GATOR2) complex inhibits the Gap activity towards Rags 1 (GATOR1), prompting activation of the mTOR complex 1 (mTORC1) which induces a signaling cascade that positively regulates cell growth, metabolism, and autophagy (reviewed in ref. 10,11). Conversely, metabolic stress signals like hypoxia prompt activation of the tuberous sclerosis 1 (TSC1) and 2 (TSC2) complex, which inhibits mTORC1 signaling by inactivation of Ras homolog, mTORC1 binding (Rheb) (reviewed in ref. 10).

Notably, dysregulation of mTORC1 signaling in oocytes has been shown to have varying impacts on short- and long-term fertility across organisms through poorly understood mechanisms[12–17]. For example, in *Drosophila* ovarioles with reduced mTORC1 signaling due to knockout of the GATOR2 complex protein, Missing oocyte (*mio*; *mios* in zebrafish), oocytes are initially specified but not maintained[18]. How mTORC1 signaling contributes to sustained oogenesis remains unclear. In other cell types, mTORC1 signaling has been shown to support early stages of ribosome biogenesis[19]. Sufficient ribosome biogenesis in oocytes is vital to support their translational burden and to provide the ribosomes the embryo requires (reviewed in ref. 4). However, the interplay between nutrition, mTORC1 signaling, and ribosome biogenesis in the context of oocyte development remains to be fully understood.

Here we identified a role for the vertebrate-specific oocyte fate regulator Rbpms2 in nucleolar expansion and mTorc1 signaling. Using an oocyte-specific tagged-Rbpms2 for RNA immunoprecipitation, we found that Rbpms2 target RNAs in oocytes *(Rbpms-bound oocyte RNAs; rboRNAs)* include regulators of RNA metabolism and degradation, regulators of ribosome biogenesis, and surprisingly, RNAs required for testis development. Cross-referencing the *rboRNAs* to previously published single-cell RNA-sequencing ovary data[20], we identified an enrichment of *rboRNAs* in early oocytes, the stages crucial for further oocyte and ovary differentiation in zebrafish. Subsequent RNA-seq analysis implicates Rbpms2 as a *rboRNA* translational regulator, likely acting to repress testis factors and activate ovary factors. Further, RNA-seq and immunohistological analyses indicate that Rbpms2 promotes nucleolar assembly, ribosome biogenesis, and mTorc1 pathway components. Thus, we hypothesized that early gonocytes are in a dual poised bipotential state where Rbpms2 dynamically regulates oocyte and testis *rboRNAs* and acts as a switch upstream of a nutrient-responsive pathway, positively impacting the translational capacity of oocytes.

Accordingly, genetic analyses place Rbpms2 upstream of the mTorc1 pathway-activating protein, Mios. Furthermore, Mios loss, like loss of Rbpms2, disrupts nucleolar composition and compromises ovary maintenance. Genetically manipulating regulators of the metabolic stress sensing arm of mTorc1 in *mios* mutants (*mios*[−/−]) did not consistently prevent oocyte loss, while active forms of mTOR restored oogenesis and female sex determination in *mios*[−/−] fish. Cumulatively, we have identified an RNA binding protein (RNAbp)-mediated binary fate switch that activates a Gator2-mediated checkpoint essential for oocyte development, thereby integrating sexual differentiation factors and nutritional availability pathways during zebrafish oogenesis and female sex determination.

## Results

### Rbpms2 binds to testis development factors in early oocytes

In nongonadal somatic cells, Rbpms family proteins are thought to act primarily as translational regulators. Recently this has been shown during cardiac commitment of human embryonic stem cells (hESCs) where RBPMS controls the translation initiation of several factors, and subsequent developmental pathways, by recruitment to actively translating ribosomes[21]. However, the germline-specific function(s) and RNA target(s) of Rbpms2 have yet to be characterized[22]. Therefore, we sought to determine the germline-specific RNA targets of Rbpms2, which are expected to be critical for successful oogenesis and reproductive functions.

As Rbpms2 is expressed in multiple tissues[2], we took a transgenic RNA immunoprecipitation (RNAIP) approach to isolate Rbpms2 target RNAs in oocytes. Briefly, we used previously generated stable adult transgenic fish expressing mApple-Rbpms2 or mApple alone under the early oocyte-specific promoter, *buckyball* (*buc;* Fig. 1a)[2,23]. This promoter was selected because it is specifically expressed in oocytes but not ovarian somatic cells[23,24], *buc* is present in the same cell types of the early ovary as *rbpms2a* and *rbpms2b* (Supp. Figure 1), and previous work in our lab has shown that these proteins both localize to the Balbiani body of early oocytes and are not present in any cells of the testis[2,3]. We identified 732 RNAs bound by Rbpms2 in oocytes of adult ovaries (*rboRNAs*) (Fig. 1b). Among *rboRNAs*, 149 have been associated with spermatogenesis and testis fate (*rbtRNAs*) in zebrafish and other organisms (Fig. 1b and Supp. Fig. 1a).

Mapping the temporal expression of *rboRNAs* to previously generated 40 days post fertilization (dpf) zebrafish ovary single-cell RNA-sequencing data[20] revealed that 725 of these RNAs are expressed throughout mitotic, meiotic, and early oocyte cells (Fig. 1b). Of the 7 unmapped *rboRNAs*, 5 transcripts are expressed in later stage oocytes[25,26] present in the fully mature adult ovary but not in the 40 dpf ovary dataset. The remaining 2 have yet to be characterized in oocytes. Notably, several *rbtRNAs* were enriched in the undifferentiated mitotic and pre-meiotic cells of the 40 dpf zebrafish ovary (Fig. 1b)[20]. This is consistent with our previous findings that *rbpms2a; rbpms2b* double mutant (*rbpms2* DMs) zebrafish initiate testis development in response to disrupted oogenesis progression[2]. Therefore, we hypothesize that Rbpms2 negatively regulates several *rbtRNAs* in gonocytes to promote the early stages of oogenesis and female sex differentiation.

Our previous work has shown that Rbpms2 is not only required to suppress testis fates but must also promote factors and pathways related to ovary fates for successful oogenesis[3]. To better understand the pathways by which Rbpms2 may promote ovary fates specifically, we performed GO Term Biological Process analysis on *rboRNAs* (Fig. 1c). Of the Top 10 terms, several *rboRNAs* were related to mRNA metabolism and processing and ribosome biogenesis and ribonucleoprotein complex biogenesis and assembly. It has been established in invertebrates and vertebrates that a sufficient ribosomal quantity, and therefore translational capacity, is vital for productive oogenesis and subsequent embryogenesis (refs. 27,28, reviewed in ref. 4). This suggests that Rbpms2 may positively regulate *rboRNAs* related to ribosome biogenesis and ribonucleoprotein biogenesis and assembly to ensure sufficient levels of each are present for oogenesis.

Several mechanisms contribute to cell fate decisions, such as regulation of RNA biogenesis and metabolism, including RNA stabilization and decay which are known functions of RNAbps (reviewed in ref. 29), and are important for oogenesis[30–32]. To determine if Rbpms2 regulates *rboRNA* abundance, we performed bulk RNA sequencing on *rbpms2* DMs and wildtype fish at 21 dpf, prior to sex determination, and evaluated global RNA and *rboRNA* differential expression. Of 14257 global RNAs present, only 2% were significantly downregulated and 7% upregulated between wildtype and *rbpms2* DMs (Fig. 1d). Overlaying *rboRNAs* on this dataset showed that 728/730 targets are not significantly changed (Fig. 1d). This suggests that Rbpms2 does not regulate RNA abundance but could instead function at the level of translational control. These findings are consistent with the following hypotheses: (1) Rbpms2 likely functions to repress translation of *rbtRNAs* expressed in early gonocytes in ovaries and (2) promote

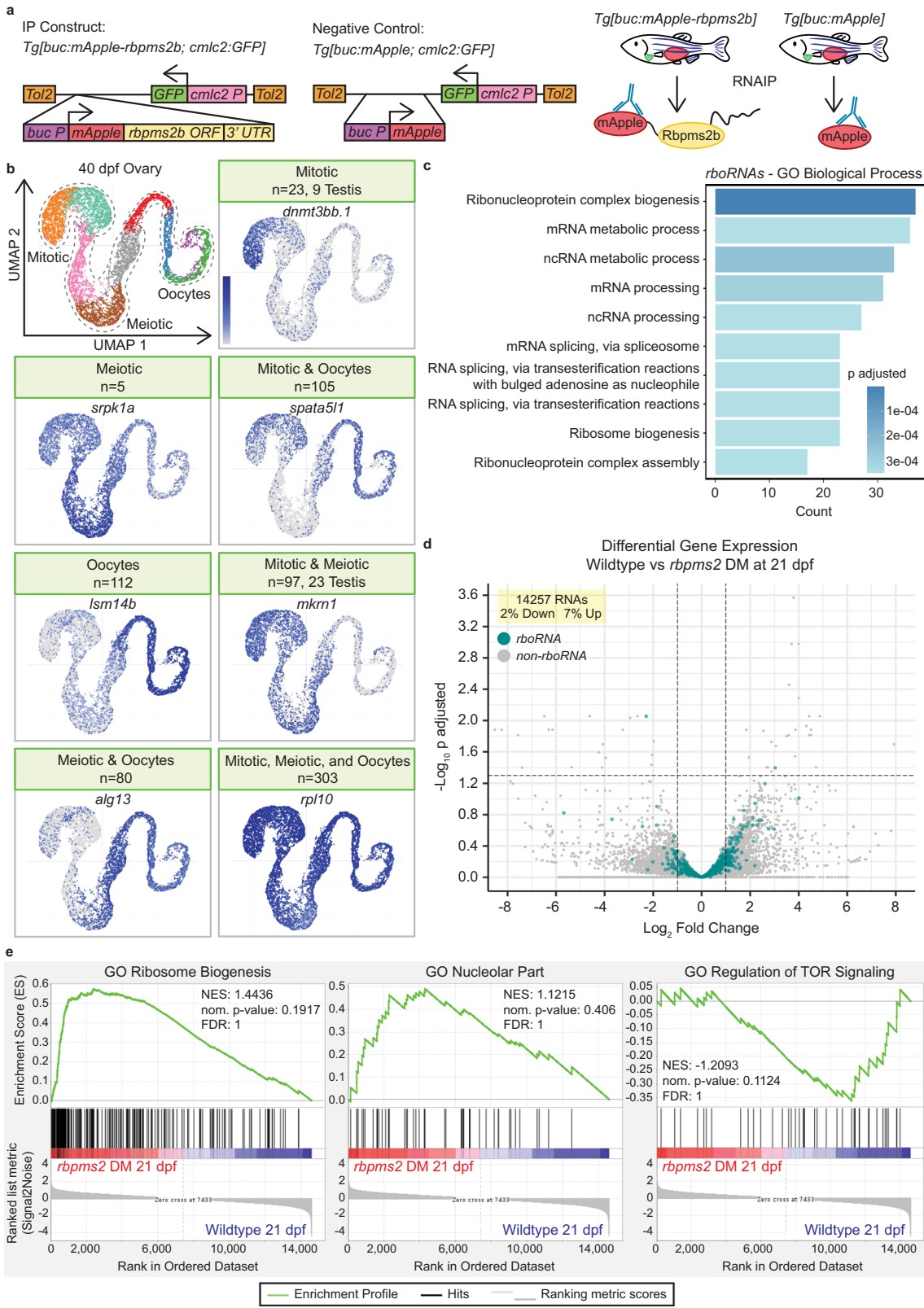

translation of oocyte factors, including ribosome and ribonucleoprotein complex biogenesis regulators to support growth, oocyte differentiation, and meiotic progression, though alternative mechanisms cannot be excluded.

To further understand the pathways by which Rbpms2 promotes oogenesis and female sex differentiation, we analyzed Gene Set Enrichment (GSEA) plots from our bulk RNAseq dataset (Fig. 1e). These plots show the enrichment scores of RNAs related to specific GO terms between genotypes and reveal that RNAs related to ribosome biogenesis and the nucleolus, the hub of ribosome biogenesis, were highly enriched in *rbpms2* DMs as compared to wildtype fish. This further suggests that Rbpms2 likely functions at the level of translation to promote ribosome biogenesis, a process essential to oogenesis, and that the absence of Rbpms2 may disrupt a feedback loop between

**Fig. 1 | Rbpms2 binds and regulates RNAs required for testis development and ribosome and ribonucleoprotein biogenesis. a** RNAIP and Negative control constructs and RNAIP scheme. **b** UMAP of cell populations in 40 dpf ovary[20] and representative UMAPs of *rboRNAs* in given cell population(s) of the 40 dpf ovary. Number of RNAs in a category are given in the header and *rboRNAs* associated with testis fates are specified in the header as well. **c** Top 10 GO biological processes associated with *rboRNAs*. Dark blue indicates not significant and increasing lighter shades of blue indicate increasing significant values. **d** Volcano plot of differential gene expression in 21 dpf bipotential wildtype and *rbpms2* DM gonads from bulk RNA sequencing. *rboRNAs* are overlayed in blue and all RNAs are labeled in gray. (c-

d) Differential Gene Enrichment was calculated with the DESeq2 package adjusted $P = 0.05$, adjusted $P$ ($-\log_{10}$) = 1.30 and fold change of 2.00, $\log_2$ fold change ($-1.00|$ 1.00). **e** Representative gene set enrichment plots for GO Ribosome Biogenesis, GO Nucleolar Part, and GO Regulation of TOR Signaling from bulk RNA sequencing of 21 dpf wildtype versus *rbpms2* DM gonads. Dark red indicates more frequently detected in *rbpms2* DM gonads, pink indicates slightly more expressed in *rbpms2* DM gonads, light blue indicates slightly more expressed in wildtype gonads, and dark blue indicates more frequently expressed in wildtype gonads. NES, nominal p-value, and FDR were calculated via the Gene Set Enrichment Analysis algorithm. Source data are provided as a Source Data file.

ribosome biogenesis, RNA transcription, and translation. Further, expression of RNAs related to regulation of TOR signaling are greatly decreased in *rbpms2* DMs at 21 dpf as compared to wildtype (Fig. 1e). mTOR signaling, specifically via the mTORC1 nutrient-sensing branch, is a known regulator of nucleolar formation and all stages of ribosome biogenesis (reviewed in ref. 33). These results suggest that early gonocytes exist in a bipotential state wherein Rbpms2 mediates a binary fate-switch likely by indirectly or directly repressing *rbtRNA* translation and promoting nucleolar formation and oocyte development via mTor signaling to ensure sufficient ribosome biogenesis for oogenesis.

## Rbpms2 regulates nucleolar amplification

Prophase I is a protracted phase of meiosis that occurs in early oocytes, and is a period of significant transcription prior to diplotene arrest (reviewed in ref. 34). During meiotic progression, oocytes cease transcription and ramp up translation to support major oocyte growth, prepare maternal reserves, and support the translational requirements of the future embryo. This translational demand requires a large pool of translational machinery, most notably the ribosome. Ribosome biogenesis occurs in the nucleolus, a structure composed of proteins, ribosomal DNA (rDNA), and ribosomal RNA (rRNA) assembled around nucleolar organizer regions into distinct compartments (reviewed in ref. 35). To support the high generation of ribosomes, teleosts and other fish and amphibian oocytes amplify their single nucleolus to multiple nucleoli in pachytene with nucleolar expansion and maturation culminating at diplotene when the oocyte arrests in meiosis I (Fig. 2a)[36-38].

Given the abundance of nucleoli and ribosome biogenesis-related *rboRNAs*, we reanalyzed transmission electron microscopy (TEM) images of *rbpms2* DM oocytes[2] to assess nucleolar features. Ultrastructural analyses and quantification of nucleoli number revealed that wildtype prophase I oocytes had greater than four nucleoli per cell at 35 dpf (Fig. 2b, d) whereas most *rbpms2* DM oocytes had fewer than four nucleoli (Fig. 2c, d).

Nucleoli are composed of distinct compartments that regulate rRNA transcription, modification, or early ribosomal protein assembly (reviewed in ref. 35). To determine if nucleolar development requires Rbpms2, we analyzed markers of distinct ribosome biogenesis stages in wildtype and *rbpms2* DMs. Prior work has established three phases of nucleolar development: a nucleation phase mediated by rDNA, a seeding phase characterized by recruitment of RNA polymerase I (RNA pol I), which is required for rRNA transcription and functions within the nucleolus, and a growth limited phase[39]. To examine nucleation and visualize RNA pol I activity, we stained for phosphorylated Upstream binding transcription factor (p-Ubtf), which is required for active RNA pol I. In wild-type ovaries, RNA pol I nuclear localization was exclusive to nucleoli, as expected, in all germ cells (Fig. 2e). We also observed localization of RNA pol I to the cytoplasm of oocytes and early meiotic cells.

To compare rRNA and mRNA transcription in wild-type oocytes, we assessed the distribution of RNA polymerase II (RNA pol II) and RNA pol I. In wild-type gonads, we observed RNA pol II in all mitotic and

early meiotic germ cell nuclei in addition to some somatic cell nuclei. As expected, RNA pol II was strongly localized to oocyte nuclei and did not overlap with RNA pol I, which is restricted to nucleoli within the nucleus (Fig. 2e and Supp. Fig. 2). In *rbpms2* DMs, while RNA pol I localized to the cytoplasm and nucleoli of mitotic germ cells, it was observed in oocyte nucleoli as well as throughout nuclei, overlapping with RNA pol II nuclear localization (Fig. 2f). This finding indicates that recruitment of RNA pol I is compromised and suggests that rDNA nucleation and/or nucleolar seeding by rRNA transcription may be disrupted in *rbpms2* DM oocytes.

To further characterize nucleoli, we analyzed Fibrillarin, a conserved nucleolar small nuclear ribonucleoprotein component that localizes to nucleoli[40]. In wildtype ovaries, Fibrillarin localized to the single nucleolus of mitotic germ cells and was present in the nucleoli of oocytes of all stages (Fig. 2g). In *rbpms2* DM oocytes, Fibrillarin was localized to a few, small nucleoli or dispersed throughout the nucleus with no specific localization (Fig. 2h). Notably, nucleolar localization of RNA pol I and Fibrillarin were consistent with ultrastructural data indicating few nucleoli per oocyte in *rbpms2* DM ovaries. Taken together, these results indicate that nucleolar recruitment of ribosome biogenesis factors is dysregulated in *rbpms2* DM oocytes and that Rbpms2 contributes to ribosome biogenesis and may do so by regulating nucleation and/or seeding by regulating translation of ribosome biogenesis factors that promote nucleolar development in primary oocytes.

## Rbpms2 functions upstream of the mTorc1 regulator, Mios

An essential pathway regulating nucleolar formation and ribosome biogenesis is the nutrient-sensing mTorc1 pathway[41]. Due to impaired nucleolar expansion in *rbpms2* DMs (Fig. 2b–d) and dysregulated recruitment of ribosome biogenesis factors to nucleoli (Fig. 2e–h), we determined if this phenotype was mTorc1-related. In wild-type ovaries, localization of phosphorylated p70-S6K (p-Ps6k; present as Rps6kb1a and Rpskb1b in zebrafish), a kinase directly phosphorylated by the active form of mTorc1 (reviewed in ref. 11), was asymmetrically distributed in mitotic nuclei, diffusely localized in early meiotic cell nuclei, and shifted to the cytoplasm of oocytes (Fig. 3a, c and Supp. Fig. 3a). In *rbpms2* DMs, while p-Ps6k localization in mitotic cells and gonocytes was comparable to wildtype, p-Ps6k was significantly decreased in DM oocytes (Fig. 3b, d, e and Supp. Figs. 3b and 4a, b). This observation suggests that the absence of Rbpms2 results in dysregulated mTorc1 signaling that may be a consequence of, or in turn result in, impaired ribosome biogenesis and subsequent oogenesis deficits.

To investigate the relationship between Rbpms2 and mTorc1, we evaluated potential Rbpms2 targets related to amino acid-sensing and energy deficiency regulators of the mTorc1 signaling pathway (Fig. 3f). Notably, the Gator2 complex protein, Missing oocyte (Mios), contains 4 Rbpms2 binding sites in its 3' UTR and has been previously shown to be dispensable for viability but required for oogenesis in *Drosophila*[18,42]. *Mios* transcripts were detected in mitotic and meiotic germ cells, including oocytes, of the 40 dpf ovary and was not differentially expressed between wildtype and *rbpms2* DMs at 21 dpf (Supp. Fig. 5a, b). However, Mios protein was detected in wildtype oocytes (Fig. 3g–i) and was significantly decreased in *rbpms2* DM oocytes

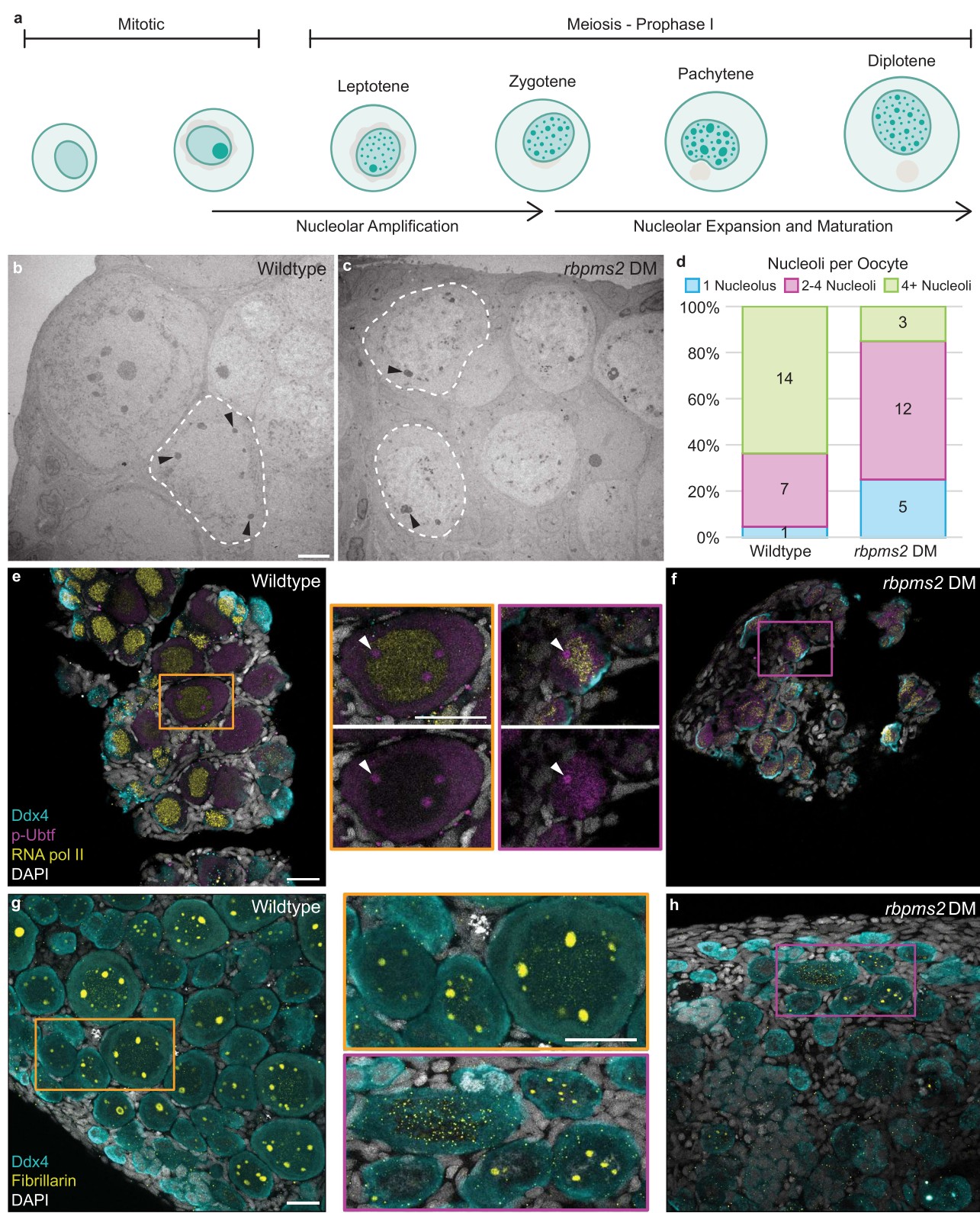

(Fig. 3j–m and Supp. Fig. 4c, d). Together, these results suggest that Rbpms2 is a positive regulator of *mios* RNA translation and therefore acts upstream of mTorc1 signaling.

## Mios is required for nucleolar development and sustained oogenesis

To investigate the contribution of Mios to sex-specific germ cell differentiation in zebrafish, we obtained an ENU-induced T > A point

mutation from the Sanger Institute Mutation project, *mios^sa22946*, which causes a premature stop codon (93L-> stop) within exon 2 (Fig. 4a)[43]. We confirmed the *mios^sa22946* mutation by sequencing both genomic DNA and cDNA and developed restriction digest based genotyping assays to distinguish the wildtype and mutant alleles (Fig. 4a). In addition, we used CRISPR/Cas9 mutagenesis to generate additional mutant alleles disrupting exon 2 of zebrafish *mios*[44]. Several in-frame deletions and nonsense mutations were recovered including *mios^ms20*,

**Fig. 2 | Nucleolar and ribosome biogenesis factors are dysregulated in *rbpms2* DMs. a** Schematic of nucleolar development from mitosis to prophase I arrest. Briefly, upon entry into meiosis, the singular nucleolus of the germ cell disassembles. As the cell progresses through prophase I, new nucleoli are amplified through seeding by rDNA loci and nucleoli begin to expand. Nucleolar expansion, maturation, and ribosome biogenesis continues up until meiotic arrest at diplotene. **b, c** Transmission electron micrographs of 35 dpf wildtype and *rbpms2* DM oocytes. Dashed lines outline oocyte cytoplasmic membrane and black arrowheads indicate nucleoli. Scale bar is 5 μM. **d** Quantification of number of nucleoli per oocyte in wildtype (*n* = 2 fish) and *rbpms2* DM (*n* = 2 fish) gonads. A minimum of 20 prophase I cells per genotype were analyzed and categorized into the following groups: 1 nucleolus (blue), 2–4 nucleoli (purple), and 4+ nucleoli (green). Source data are provided as a Source Data file. **e, f** p-Ubtf (active RNA pol I; purple) and RNA pol II (yellow) immunostaining in 31-35 dpf wildtype (*rbpms2a^{ae30}; rbpms2b^{sa9329}* heterozygous mutant; *n* = 4) and *rbpms2* DM (*rbpms2a^{ae30}; rbpms2b^{sa9329}* double homozygous mutant; *n* = 4) germ cells. Enlarged views of boxed regions show representative localization of p-Ubtf in wildtype and *rbpms2* DM oocytes and white arrows indicate p-Ubtf localization in nucleoli. Scale bar for all images is 20 μM. **g, h** Fibrillarin (yellow) immunostaining in 29-33 dpf wildtype (*rbpms2a^{ae30}; rbpms2b^{sa9329}* heterozygous mutant; *n* = 4) and *rbpms2* DM (*rbpms2a^{ae30}; rbpms2b^{sa9329}* double homozygous mutant; *n* = 4) germ cells. Enlarged views of boxed regions show representative localization of Fibrillarin in wildtype and *rbpms2* DM oocytes. Scale bar for all images is 20 μM. **e–h** Germ cells are labeled by Ddx4 (teal) and nuclei are labeled with DAPI (white).

an 11 bp deletion allele that leads to a frameshift and premature stop codon upstream of all Mios functional domains (Fig. 4a). We confirmed that Mios protein is absent in *mios^{sa22946}* mutant gonads; therefore, *mios^{sa22946}* is likely a null allele (Supp. Fig. 5c).

Analysis of *mios^{ms20}* and *mios^{sa22946}* single heterozygous and mutant progeny showed they are viable to adulthood with no overt morphological deficits. However, homozygous mutants for *mios^{sa22946}* grew slower than their heterozygous siblings, consistent with a deficit in mTorc1 signaling (Supp. Fig. 5d). Additionally, gonad morphological and sex ratio analyses show that *mios^{ms20}* and *mios^{sa22946}* mutants and *mios^{ms20/sa22946}* compound heterozygotes develop functional testes and differentiate as males, exclusively (Fig. 4b–e). As *mios^{ms20}* and *mios^{sa22946}* gonad morphology and differentiation were indistinguishable, both alleles are likely protein null, and all further analyses were performed on *mios^{sa22946}* fish (hereafter *mios^{-/-}*). Investigation of timepoints prior to testis differentiation, specifically 35 dpf, showed that *mios^{-/-}* do initially undergo early oogenesis. However, mutant oocytes remain small, and do not make it past diplotene, resulting in subsequent testis development (Fig. 4c, g).

Analysis of Fibrillarin in wildtype gonads revealed nucleolar amplification as expected (Fig. 4f). In contrast, *mios^{-/-}* oocytes contained a similar number of Fibrillarin puncta, but these were significantly smaller (Fig. 4g, h and Supp. Fig. 4e, f). Unlike Fibrillarin, RNA pol I showed no significant differences in nucleolar localization between wildtype and *mios^{-/-}* oocytes (Fig. 4g, i). In wildtype, exclusion of RNA pol II from nucleoli revealed a nucleolar region devoid of RNA pol I (Fig. 4i). This is consistent with the localization of ribosome biogenesis factors to distinct compartments of nucleoli (reviewed in ref. 35). In *mios^{-/-}* oocytes, the nucleolar region devoid of RNA pol I was not observed (Fig. 4j). This observation together with Fibrillarin staining indicates the nucleoli of *mios^{-/-}* may be immature or have impaired recruitment of certain ribosome biogenesis factors.

To investigate nucleolar architecture, we performed ultrastructural analysis of wildtype and *mios^{-/-}* oocytes (Supp. Fig. 6a, b). We did not observe significant differences between nucleoli number, morphology, or size in wildtype and *mios^{-/-}* oocytes (Supp. Fig. 6a–d). This suggests that overt nucleolar architecture is not disrupted; however, nucleolar function, recruitment, or retention of specific factors may be impaired in the absence of Mios. Taken altogether, we conclude that impaired ovary differentiation is due to mutation of *mios* and that Mios, like Rbpms2, is required for oocyte differentiation in zebrafish.

**Mios promotes oogenesis through mTorc1**

Next, we determined if Mios' functions in oogenesis and female sex differentiation are mediated through mTorc1 signaling in ovaries. P-Ps6k localization in *mios^{-/-}* was similar to early cells of wildtype gonads, however p-Ps6k in *mios^{-/-}* oocytes was significantly reduced (Fig. 5a–e and Supp. Figs. 3c, d and 4g, h). To determine if activation of mTorc1 could suppress the *mios^{-/-}* phenotype, we generated transgenic lines expressing constitutively active human L1460P mTOR (*mTOR^{ca}*)

under the germline-specific promoter, *ziwi*[45] (Fig. 5f). mTOR^{ca} has been shown to specifically increase mTORC1 downstream targets in HEK-293 cells[46]. We generated two alleles representing independent insertion events, *ms49* and *ms64*, of the *mTOR^{ca}* transgene. Neither allele disrupted oogenesis or spermatogenesis in wildtype genotypes, and both restored female development in *mios^{-/-}*, albeit to varying extents (Fig. 5g). Specifically, the *ms49* allele drove a female bias in wildtype and heterozygotes and reestablished typical male and female sex ratios in *mios^{-/-}* fish. The *ms64* allele did not bias sex development of wildtype and heterozygous fish from an expected 50:50 ratio and was less efficient at restoring oogenesis in *mios^{-/-}* fish.

To determine the fertility of *mios^{-/-}* females expressing the *mTOR^{ca}* transgene, fertility assays were performed. Briefly, paired matings of individual female and males that were transgenic positive or non-transgenic, negative controls of homozygous *mios* mutant, homozygous wildtype, and heterozygous genotypes, were conducted to determine fertility rates and determine egg quality. *mios^{-/-}* females expressing the *ms49* and *ms64 mTOR^{ca}* transgenes had comparable fertility rates to their wildtype and heterozygous transgenic and non-transgenic female siblings (Fig. 5h). However, *ms49* transgenic *mios^{-/-}* females produced significantly more total eggs than wildtype non-transgenic siblings and significantly more degenerating eggs than all transgenic and non-transgenic siblings (Fig. 5h and Supp. Fig. 7a–d).

While *ms64* transgenic *mios^{-/-}* females did not differ from their transgenic and non-transgenic siblings in number of degenerating eggs produced, they laid a significantly higher number of total eggs as compared to their transgenic wildtype siblings (Fig. 5h). These fish also produced several eggs with egg activation deficits (Supp. Fig. 7e–h). These findings indicate incomplete suppression of oocyte loss or that Mios may have mTorc1 independent roles. These results demonstrate that the requirement for Mios in oogenesis is conserved. Further, in ovaries Mios acts through mTorc1 signaling and mTorc1 activation is sufficient for successful oogenesis in the absence of Mios.

**Mios functions independent of double strand break repair**

In *Drosophila* ovaries, GATOR1 promotes meiotic entry of ovarian cysts by downregulating mTORC1 activity, which is subsequently reinforced by double-strand breaks (DSBs)[18,47]. Accordingly, in *Drosophila* mutants for Mios (*mio*), removal of GATOR1 activity or blocking DSB formation leads to elevated mTORC1 activity and restores oocyte development and fertility[18,42,47]. Spo11 (SPO11 initiator of meiotic double stranded breaks) is a conserved, essential enzyme that generates meiotic DSBs[48–50]. In zebrafish, Spo11 is not required for viability or oocyte production in females, although mutant gametes are likely aneuploid as embryos of mutant mothers are not viable[50].

To investigate if coordination between DSB repair and oocyte growth and progression mediated by mTorc1 signaling is a conserved feature of oogenesis, we generated double mutants lacking *mios* and *spo11* (*spo11^{uc73}*)[50]. As expected, based on *mios'* germline specific functions, we saw no suppression or worsening of the *mios^{-/-}* growth defect in the absence of *spo11* (Supp. Fig. 5d). Although adult male and

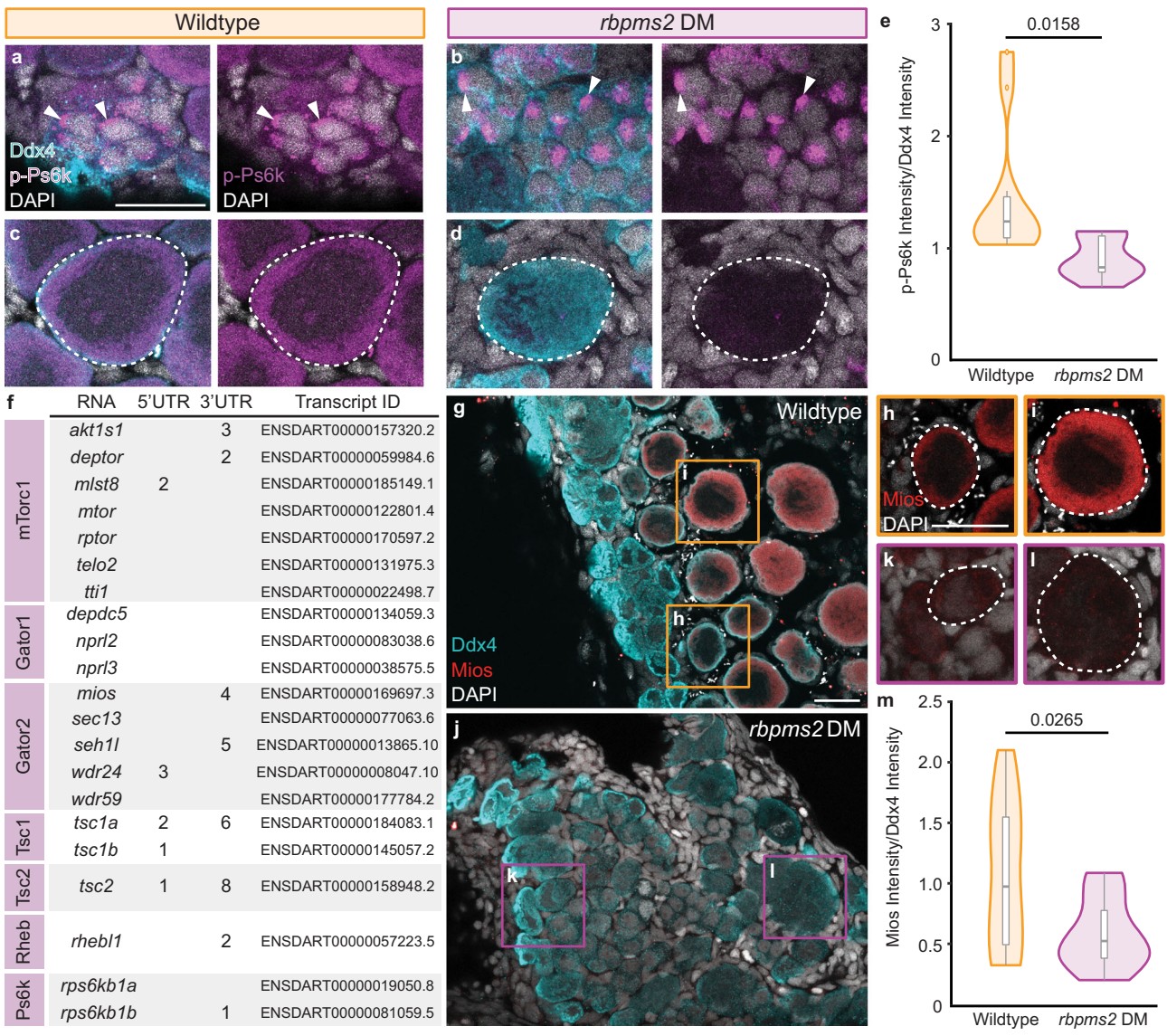

**Fig. 3 | Rbpms2 functions upstream of the Gator2 complex protein, Mios.**
**a–d** Immunostaining of p-Ps6k (purple) in 29-33 dpf wildtype (*rbpms2a^ae30^; rbpms2b^sa9329^* heterozygous mutant; *n* = 4) and *rbpms2* DM (*rbpms2a^ae30^; rbpms2b^sa9329^* double homozygous mutant; *n* = 4) germ cells. Germ cells are labeled with Ddx4 (teal) and nuclei are labeled with DAPI (white). **a, b** Arrows indicate p-Ps6k localization in mitotic and early meiotic nuclei. **c, d** Dashed lines indicate early oocytes. Adjacent panel shows p-Ps6k and DAPI localization in indicated cells. Scale bar is 20 μM. **e** Quantification of the p-Ps6k intensity normalized to the intensity of Ddx4 for wildtype (*rbpms2a^ae30^; rbpms2b^sa9329^* heterozygous mutant; *n* = 10 cells from 3 individual fish) and *rbpms2* DM (*rbpms2a^ae30^; rbpms2b^sa9329^* homozygous mutant; *n* = 9 cells from 3 individual fish) oocytes. Two-tailed paired equal variance Student's *t* tests were performed for the indicated groups. *P* = 0.05. **f** Table of mTorc1-related components with predicted Rbpms binding sites[22] in their 3′ and/or 5′ UTRs. Transcript IDs evaluated for each component are listed and purple boxes indicate the protein component or complex of the pathway a given

factor belongs to. **g–l** Immunostaining for Mios (red) in 29-31 dpf wildtype (*rbpms2a^ae30^; rbpms2b^sa9329^* heterozygous mutant, *n* = 4) and *rbpms2* DM (*rbpms2a^ae30^; rbpms2b^sa9329^* double homozygous mutant; *n* = 4) germ cells. Germ cells are labeled with Ddx4 (teal) and nuclei are labeled with DAPI (white). **h–l** Dashed outline indicates oocytes and shows Mios and DAPI localization in indicated cells. Scale bar is 20 μM. **m** Quantification of the intensity of Mios normalized to the intensity of Ddx4 for wildtype (*rbpms2a^ae30^; rbpms2b^sa9329^* heterozygous mutant; *n* = 11 cells from 3 individual fish) and *rbpms2* DM (*rbpms2a^ae30^; rbpms2b^sa9329^* double homozygous mutant; *n* = 11 cells from 3 individual fish) oocytes. Two-tailed paired equal variance student's t tests were performed for the indicated groups. *P* = 0.05. **e, m** For box and whisker plots, line indicates median, boxes indicate upper (75th) and lower quartiles (25th), whiskers indicate data within 1.5 times the inner quartile range, and error bars represent minimum and maximum values. Any data outside of this range are plotted as individual points. Source data are provided as a Source Data file.

female *spo11^uc73/uc73^* were recovered, *mios^-/-^; spo11^uc73/uc73^* fish developed exclusively as males (Fig. 6a). Further, analyses of mutant gonads at 45 dpf revealed no delay or suppression of testis development in *mios^-/-^; spo11^uc73/uc73^* double mutants compared to *mios^-/-^* siblings (Fig. 6b, c). Based on these findings we conclude that eliminating DSBs is not sufficient to compensate for or bypass loss of Mios in zebrafish oocytes.

## Oocyte mTorc1 activation uniquely requires Gator2 and Mios

If our hypothesis that failed differentiation of *mios^-/-^* oocytes is solely due to constitutive downregulation of mTorc1 activity, then activating mTorc1 signaling via modulation of the insulin/stress sensing arm of the pathway may suppress the *mios^-/-^* oocyte differentiation defect as occurs in *Drosophila*[18,47]. To investigate this possibility, we crossed the *mios^sa22946^* mutant allele onto the zebrafish *TSC complex subunit 2 (tsc2;*

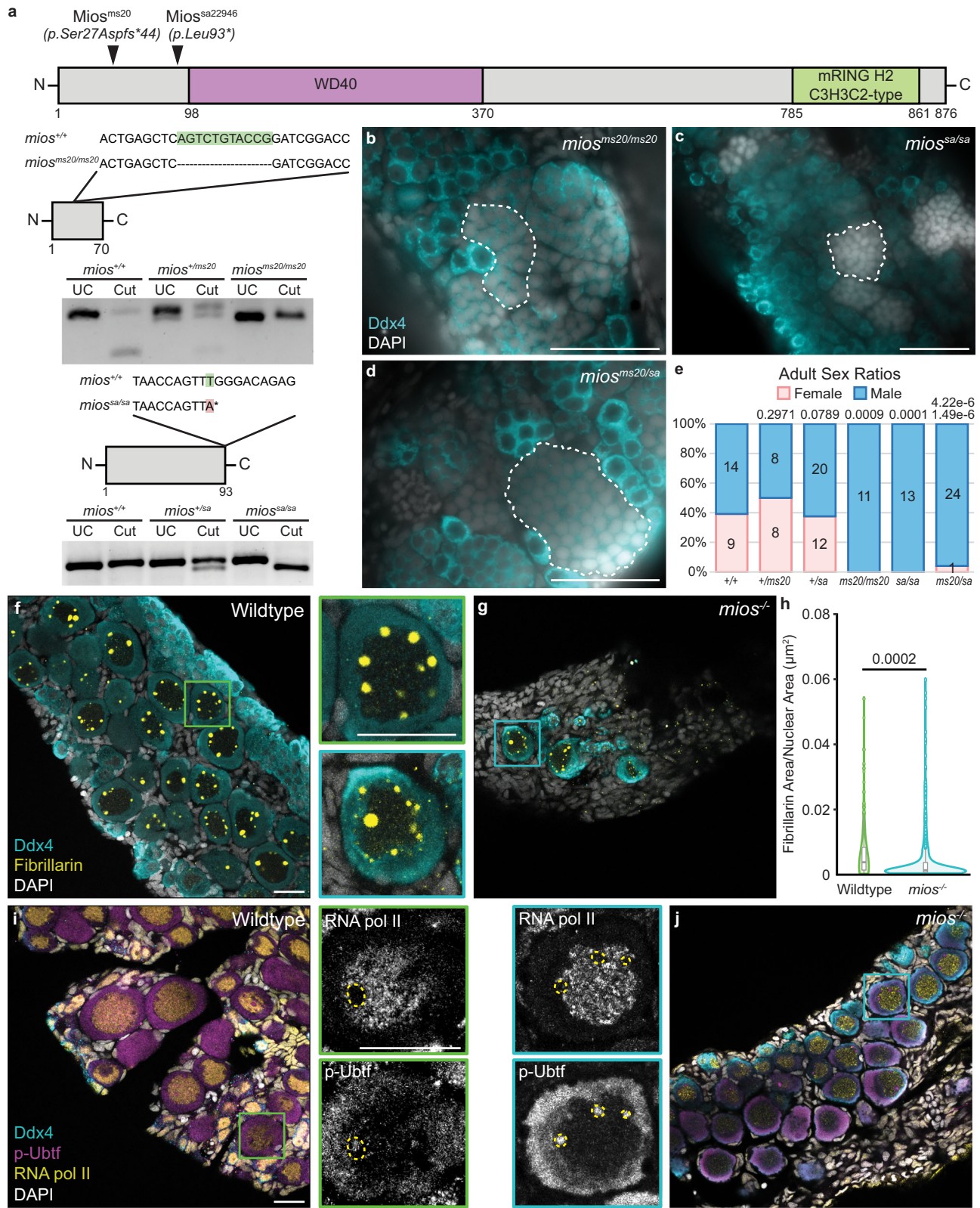

tsc2$^{vu242}$) background[51]. Tsc2 functions in complex with TSC complex subunit 1 (Tsc1) to inhibit the activity of the mTorc1 activator, Rheb (reviewed in ref. 10). Although tsc2$^{vu242/vu242}$ fish are not viable to adulthood, tsc2$^{+/vu242}$ have been shown to have elevated mTorc1[51]. Thus, if oocyte arrest is due to diminished mTorc1 in mios$^{-/-}$, then genetic reduction of tsc2 might restore sufficient mTorc1 activity to allow for oocyte progression. However, comparison of mios$^{-/-}$; tsc2$^{+/+}$ and mios$^{-/-}$;

tsc2$^{+/vu242}$ adult fish revealed no suppression of male development (Fig. 7a).

To exclude the possibility that reduction of Tsc2 did not provide sufficient mTorc1 activation, we modulated Rheb to drive mTorc1 activation through the stress sensing arm. Specifically, we generated three independent stable transgenic lines expressing constitutively active rat S16H Rheb (Rheb$^{ca}$)[52] under the germline-specific ziwi

**Fig. 4 | Mios is required for oogenesis and nucleolar maturation. a** Schematic of the Mios protein and mutagenized regions (black arrowheads) in the Mios[ms20] and Mios[sa] proteins. Truncated proteins are shown below with highlighted DNA mutations and corresponding genotyping assays. The * in Mios[sa] indicates the induced stop codon. **b**–**d** Germ cell (Ddx4, teal) and DNA (DAPI, white) in *mios[ms20/ms20]* (50 dpf; *n* = 5), *mios[sa/sa]* (45-51 dpf; *n* = 4), and *mios[ms20/sa]* (50 dpf; *n* = 4) fish. Developing sperm are outlined with dashed lines and scale bar is 50 μM. **e** Sex ratio graph for 60 dpf+ wildtype, heterozygous, and mutant fish. Pink represents female fish and blue indicates male fish. Number of fish screened are indicated for each group. Statistical significance was evaluated by performing a two-tailed $\chi^2$ test with Bonferroni correction against the +/+ control group. *P* = 0.0125. **f**, **g** Fibrillarin (yellow) immunostaining in 35 dpf wildtype (*n* = 7) and *mios[-/-]* (*n* = 10) germ cells. Enlarged views of boxed regions show representative localization of Fibrillarin in wildtype and *mios[-/-]*

oocytes. Scale bar for all images is 20 μM. **h** Quantification of the Fibrillarin puncta area normalized to oocyte nuclear area for wildtype (*n* = 9 cells from 3 individual fish) and *mios[-/-]* (*n* = 9 cells from 3 individual fish) oocytes. For box and whisker plots, line indicates median, boxes indicate upper (75th) and lower quartiles (25th), whiskers indicate data within 1.5 times the inner quartile range, and error bars represent minimum and maximum values. Any data outside of this range are plotted as individual points. Two-tailed paired equal variance Student's *t* tests were performed for the indicated groups. *P* = 0.05. **i**, **j** p-Ubtf (active RNA pol I; purple) and RNA pol II (yellow) immunostaining in 35 dpf wildtype (*n* = 4) and *mios[-/-]* (*n* = 4) germ cells. Enlarged views of boxed regions show representative localization of p-Ubtf (red) and RNA pol II (white) in wildtype and *mios[-/-]* oocytes. Scale bar for all images is 20 μM. **f**, **g**, **i**, **j** Germ cells are labeled by Ddx4 (teal) and nuclei are labeled with DAPI (white). Source data are provided as a Source Data file.

promoter[45] (Fig. 7b). Germline expression of *Rheb[ca]* did not disrupt sex differentiation in heterozygous genotypes (Fig. 7b). Although the *ms44* allele did restore oogenesis in some *mios[-/-]* fish (Fig. 7b) this effect was incompletely penetrant and was not reproducible across independent crosses, nor across independent insertions (Supp. Fig. 8). Since neither Tsc2 reduction nor Rheb[ca] reproducibly suppressed oocyte loss in *mios[-/-]*, we conclude that activating the metabolic stress arm of the mTorc1 pathway is not sufficient to support oocyte progression. Importantly, these results indicate that early oocyte development requires sufficient amino acid quantities to drive mTorc1 signaling and that oocyte progression is uniquely mediated by Gator2 and Mios.

## Discussion

Our study evaluates the mechanisms by which the vertebrate-specific RNA binding protein (RNAbp), Rbpms2, promotes successful oogenesis and female sex determination and differentiation in zebrafish. Analyses of *rboRNAs* and global RNA changes indicate mitotic and early meiotic express mRNAs encoding regulators of both sexual development programs and that Rbpms2 likely functions to translationally repress testis-associated factors and promote ovary-associated factors, including RNAs related to ribosome biogenesis (Fig. 7c). This is consistent with our previous findings that suppression of the male developmental program in *rbpms2* DMs prevents formation of a functional testis but is not sufficient to restore oogenesis[3].

We identify the mTorc1 pathway as an essential oogenesis pathway and a target of Rbpms2 activity, specifically through the Gator2 component, Mios. We find that both Rbpms2 and Mios are required for successful nucleolar development and consequently ribosome biogenesis, though they affect different stages of this process. Specifically, Rbpms2 acts early in nucleolar development, likely regulating translation of targets involved in rDNA-mediated nucleation and/or rRNA seeding of nucleoli based on the impaired recruitment of the ribosomal RNA polymerase, RNA pol I in *rbpms2* DM oocytes. Normal recruitment of RNA pol I in *mios[-/-]* suggests that Mios, which functions downstream of Rbpms2, affects the growth phase, specifically, nucleolar recruitment of pre-ribosome assembly components like Fibrillarin. Additionally, we provide genetic evidence that the relationship between DSB regulation and mTorc1 signaling observed in *Drosophila* is not conserved in zebrafish oocytes.

Finally, we demonstrate that the nutrient sensing arm of the mTorc1 pathway is uniquely required for oocyte progression and sustained oogenesis (Fig. 7c). This finding is consistent with field observations linking nutrition availability to skewed sex ratios of developing zebrafish, specifically that nutrient deprivation causes male biased development[9]. Together, these results underscore the importance of ribosome biogenesis in oocytes and suggest that oocyte development past prophase I requires a certain threshold of ribosomes. We hypothesize that if this minimum ribosome quantity is not attained in early zebrafish oocytes, oocyte development is halted and remodeling of the gonad into a functional testis is triggered.

RNAbps are known to have broad, dynamic roles in post-transcriptional regulation in several cell types across species. Our previous work has demonstrated the role of Rbpms2 in regulating testis and ovary differentiation in zebrafish, but the mechanism and its targets remained to be determined[2,3]. Through RNAIP and bulk RNA-seq experiments, we provide evidence demonstrating the essential function of Rbpms2 as a multipronged regulator of sexual differentiation. We observe few changes in overall RNA and *rboRNA* transcript abundance between wildtype and *rbpms2* DMs. Additionally, RNA pol II, which we show is present in nuclei of wildtype zebrafish oocytes up to diplotene arrest, is also intact in *rbpms2* DM oocytes. Together, these results indicate that Rbpms2 likely regulates sexual differentiation through translational control of its target RNAs as RNA pol II-mediated transcription and overall mRNA abundance appear unperturbed in *rbpms2* DM oocytes.

For example, several *rboRNAs* are associated with testis functions and their expression is limited to the early, undifferentiated cell types of the 40 dpf ovary. As the zebrafish ovary has the plasticity to switch sex and develop into testes during sex determination and differentiation (completed between 35-45 dpf) and throughout life, the presence of these transcripts in early, undifferentiated cells strongly suggest that Rbpms2 functions to repress translation of these *rbtRNAs* and prevent bipotential cells from undergoing spermatocyte development. Additionally, several *rboRNAs* have defined roles in oogenesis and ovary differentiation, including Mios shown in this work. Rbpms2 likely promotes translation of these RNAs to support oocyte differentiation. Evidence supporting this role includes absence of Mios protein in *rbpms2* DM oocytes and our previous finding that silencing of testis differentiation pathways in *rbpms2* DMs did not restore or support oogenesis[3]. Further, prior work on the related Rbpms in mouse hESCs has demonstrated that Rbpms can selectively regulate translation of its target RNAs through interactions with their 3'UTR, with some bound RNAs activated and others repressed[21]. Additionally, Rbpms can influence translational initiation independent of RNA binding through its direct binding to the translational machinery to globally promote translational initiation[21]. Further experimentation is required to determine if Rbpms2 similarly regulates *rboRNAs*, as well as investigations into its potential binding partner(s).

Active antagonism of sex differentiation pathways has been observed both in organisms that undergo polygenic and sex chromosome-mediated sex determination mechanisms (reviewed in ref. 53). For example, in female mammalian sex determination, active suppression of SOX9 by FOXL2 is required for ovary formation and differentiation during embryonic development[54]. In many species the balance of androgens and estrogen levels is crucial to proper reproductive function. In humans, dysregulation of estrogen and androgen pathways contributes to a variety of female-associated pathologies, such as premature ovarian failure where androgen levels among other things are no longer maintained at appropriate levels for typical ovarian function[55].

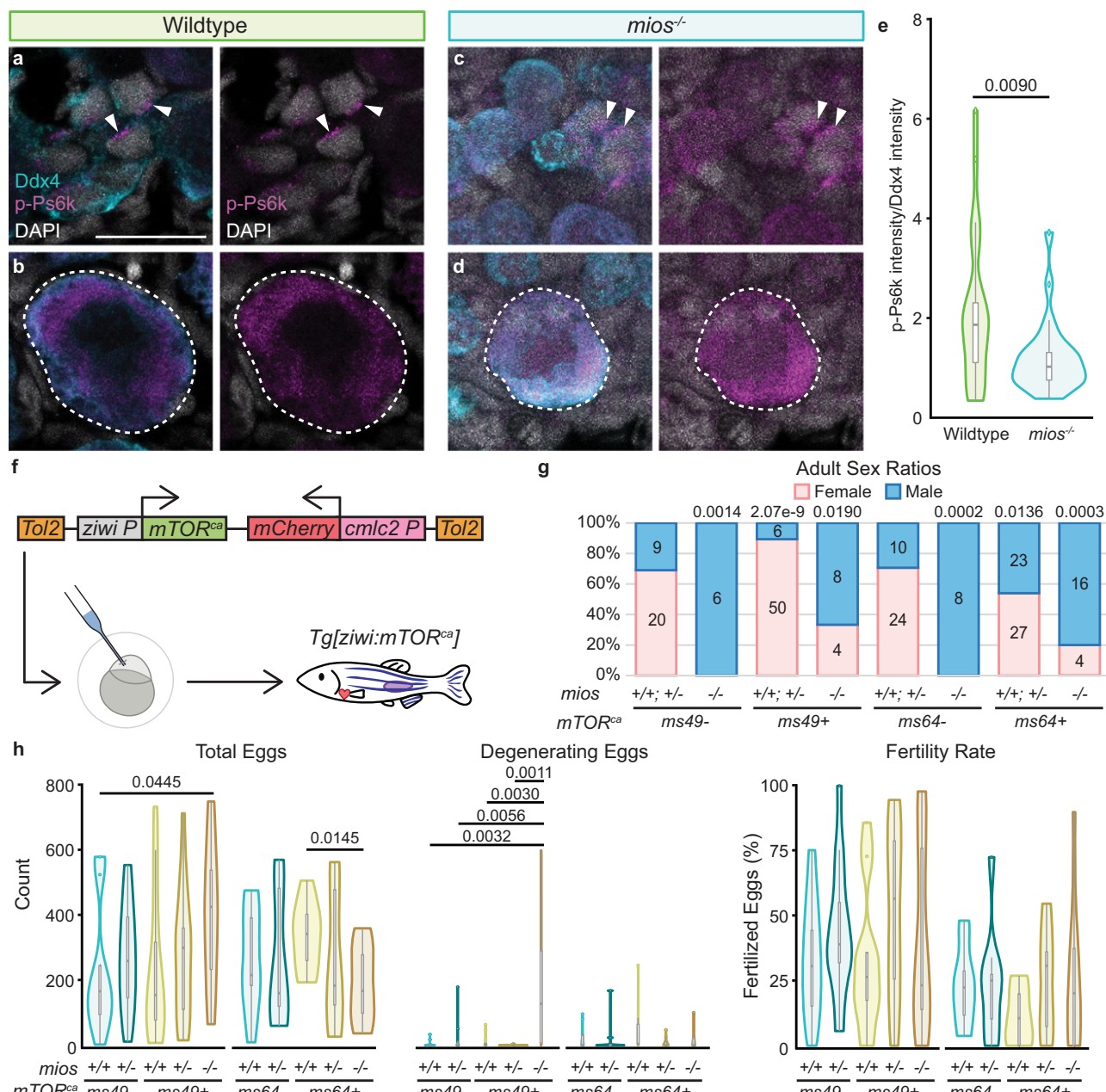

**Fig. 5 | Mios promotes oogenesis through mTorc1 signaling.**
**a**–**d** Immunostaining of p-Ps6k (purple) in 35 dpf wildtype (*n* = 7) and *mios*⁻/⁻ (*n* = 10) germ cells. Germ cells are labeled with Ddx4 (teal) and nuclei are labeled with DAPI (white). **a**, **b** Arrows indicate p-Ps6k localization in mitotic and early meiotic nuclei. **c**, **d** Dashed lines indicate early oocytes. Adjacent panel shows p-Ps6k and DAPI localization in indicated cells. Scale bars are 20 μM. **e** Quantification of the intensity of p-Ps6k normalized to the intensity of Ddx4 for wildtype (*n* = 21 cells from 7 individual fish) and *mios*⁻/⁻ (*n* = 25 cells from 8 individual fish). Two-tailed paired equal variance Student's *t* tests were performed for the indicated groups. *P* = 0.05. **f** *mTOR*ca transgene: the human *mTOR*ca expression is driven by the germline *ziwi* promoter and the heart specific promoter *cmlc2* drives *mCherry* expression as a selectable marker. Scheme depicts injection into 1-cell embryos and generation of stable *Tg[ziwi:mTOR*ca*]* lines. **g** Sex ratios for 90 dpf+ fish with *ms49* and *ms64 mTOR*ca alleles. Pink represents female and blue represents males; numbers of fish

screened are indicated for each group. Non-transgenic siblings are indicated by - and transgenic siblings are indicated with + and the specific *mTOR*ca alleles are indicated. Statistical significance was evaluated by performing a two-tailed χ² test with Bonferroni correction against the *mios +/+; +/-* non-transgenic control group. *P* = 0.0125. **h** Fertility assays and quantifications of total eggs, degenerating eggs, and fertility rates for *ms49* and *ms64 Tg[ziwi:mTOR*ca*]* fish (*n* = 3 fish per genotype and transgenic condition). Non-transgenic siblings are indicated by - and transgenic siblings are indicated with + and the specific *mTOR*ca alleles are indicated. Two-tailed paired equal variance Student's *t* tests were performed for the indicated groups. *P* = 0.05. All other comparisons were non-significant. **e**, **h** For box and whisker plots, line indicates median, boxes indicate upper (75th) and lower quartiles (25th), whiskers indicate data within 1.5 times the inner quartile range, and error bars represent minimum and maximum values. Any data outside of this range are plotted as individual points. Source data are provided as a Source Data file.

Wild zebrafish use a ZW system to determine sex[56]. Although this system was lost in lab strains, differential regulation of rDNA, coding, and noncoding loci on chromosome 4 have been linked to gonad sex in wild strains and lab strains[57–59]. Our findings position Rbpms2 as a

crucial fate switch regulator of differentiation of the bipotential zebrafish gonad, supporting oogenesis and suppressing testis development by inhibiting testis-associated RNAs and promoting oogenesis-related *rboRNAs*, affecting a wide array of downstream pathways.

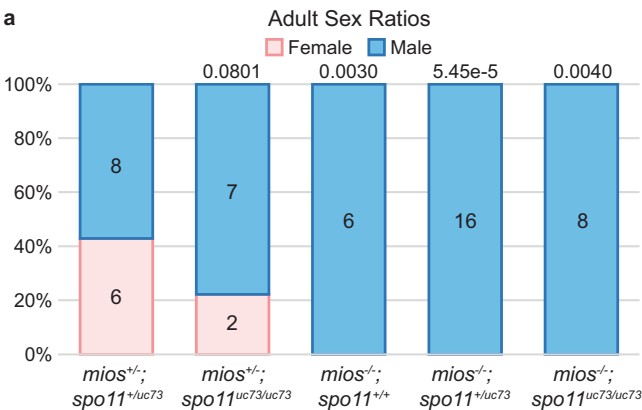

**a** Adult Sex Ratios

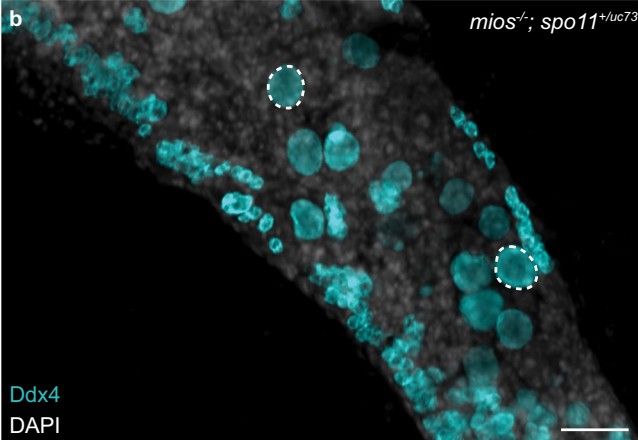

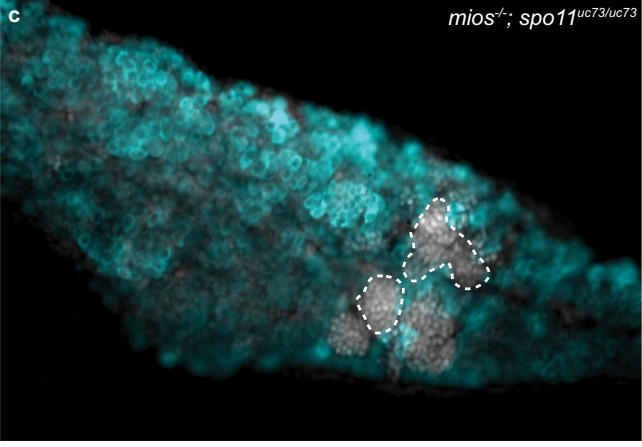

**Fig. 6 | DSB inhibition does not restore oogenesis in *mios^-/-*. a** Sex ratio graph for 60 dpf+ *mios; spo11^uc73* double heterozygous, heterozygous mutant, mutant wildtype, mutant heterozygous, and double mutant fish. Pink represents female fish and blue indicates male fish. Number of fish screened are indicated for each group. Statistical significance was evaluated by performing a two-tailed $\chi^2$ test with Bonferroni correction against the *mios^+/-; spo11^+/uw73* control group. $P = 0.0125$. Source data are provided as a Source Data file. **b** Germ cell (Ddx4, teal) and DNA (DAPI, white) in *mios^-/-* (*mios;spo11* mutant wildtype/mutant heterozygous fish, $n = 6$) and (**c**) *mios^-/-; spo11^uc73/uc73* double mutant fish ($n = 7$). Developing oocytes (**b**) and sperm (**c**) are outlined with dashed lines, respectively. Scale bar is 50 μM.

Because Mios has a conserved role in oocyte maturation in invertebrates and vertebrates (this work and[18,42,47]), the Rbpms2 and Gator2-mediated nutritional checkpoint and Rbpms2 regulation of nucleolar development identified here are expected to be required for oocyte progression and sustained ovary development even in wild zebrafish strains with an intact ZW-based chromosomal sex determination

system[60]. In these recently domesticated wild strains, heterogametic (ZW) individuals, like lab strains, form bipotential gonads that develop into juvenile ovaries that in ZW individuals usually form mature ovaries but occasionally form testes, whereas homogametic (ZZ) individuals lack juvenile ovaries and always form testis[60]. In this wild context the plasticity of the ovary and the Gator2-mediated checkpoint uncovered here could explain the observation of occasional ZW males in wild strains and WW males in lab strains, thus reconciling the notion that the W is not sufficient for ovary fate.

Nucleoli serve as the hub for rDNA amplification, rRNA splicing, and pre-ribosome assembly (reviewed in ref. 35). In oocytes, extensive amplification of the nucleolus is required to support the growth demands of the oocyte as well as prepare the necessary components passed on from the oocyte to the embryo (reviewed in ref. 61). We show that several *rboRNAs* are related to ribosome biogenesis and the nucleolus and, more broadly, that these pathways are dysregulated between undifferentiated wildtype and *rbpms2* DM fish. Further, we find that nucleolar development is dysregulated in *rbpms2* DMs and *mios^-/-* oocytes and likely contributes to failed ovary differentiation. The more severe nucleolar phenotypes and disruption of RNA pol I localization in *rbpms2* DMs but not *mios^-/-* oocytes indicates differential contributions of the proteins to nucleologenesis and is consistent with Rbpms2 functions upstream of Mios. Specifically, our results suggest that rRNA transcription and nucleoli amplification require Rbpms2 or one of its targets, whereas Mios acts later in pre-ribosomal assembly (Fig. 7c).

Transcription of rDNA has been posited to promote seeding of new nucleoli[39]. Notably, in zebrafish it has been shown that demethylation and amplification of an rDNA locus at the end of chromosome 4 (*fem rDNA*) strongly correlates with female sex determination and differentiation[62]. This suggests that oocyte development relies significantly on expansion of its ribosomal pool and positions ribosomal counting as a necessary checkpoint that must be passed for further oogenesis and female differentiation (Fig. 7c). Our results indicate the importance of Rbpms2 in ribosome biogenesis and further investigation is required to determine the relationship between Rbpms2 and rDNA amplification, like *fem rDNA*, in oocytes.

In *Drosophila*, Mio is required for oocyte progression beyond pachytene of prophase I and to maintain oocyte fate[42]. As *mio^-/-* oocytes adopt the polyploid nuclear properties of nurse cells, oocyte fate is lost, resulting in female sterility[42]. Further, Mio loss in fly oocytes alters the kinetics of synaptonemal complex formation leading to delayed or abnormal synaptonemal complex formation[42] such that oogenesis can be restored in *mio^-/-* by removing the *spo11* homolog, *mei-W86*[42,47]. As in *Drosophila*, zebrafish Mios is required in prophase I oocytes. However, unlike in flies, DSB sensing, and oocyte progression appear to be uncoupled in zebrafish, as loss of *spo11* in *mios^-/-* gonads was not sufficient to restore oocyte differentiation and maturation. Further, *spo11^-/-* zebrafish do not demonstrate a sex bias, have no overt growth defects, and progress through oogenesis and spermatogenesis (this study and[50]).

This is consistent with studies of mouse spermatogenesis, wherein inhibition of mTORC1 by conditional knockout of Raptor in the testis[63] or rapamycin exposure[64] did not disrupt DSB formation or repair. Moreover, ex vivo maturation of prematuration stage human and mouse oocytes in the presence of rapamycin results in reduction of γH2Ax, suggesting DSBs are either less numerous or are repaired faster in the context of mTOR inhibition[65]. Further evidence supporting distinct meiotic and nutritional oocyte differentiation checkpoints in vertebrates is failure of Tp53 loss to suppress oocyte loss in *rbpms2* DMs and other zebrafish mutants disrupting genes required for oocyte development through prophase I[2] and of meiotic checkpoint factors, e.g. Mei-41/ATM/ATR, to suppress loss of Mio in *Drosophila*[42]. Collectively, the available evidence suggests that in vertebrates, the meiotic checkpoint and associated DNA integrity factors that safeguard faithful replication and reduction of chromosomes as part of meiosis operates independently of

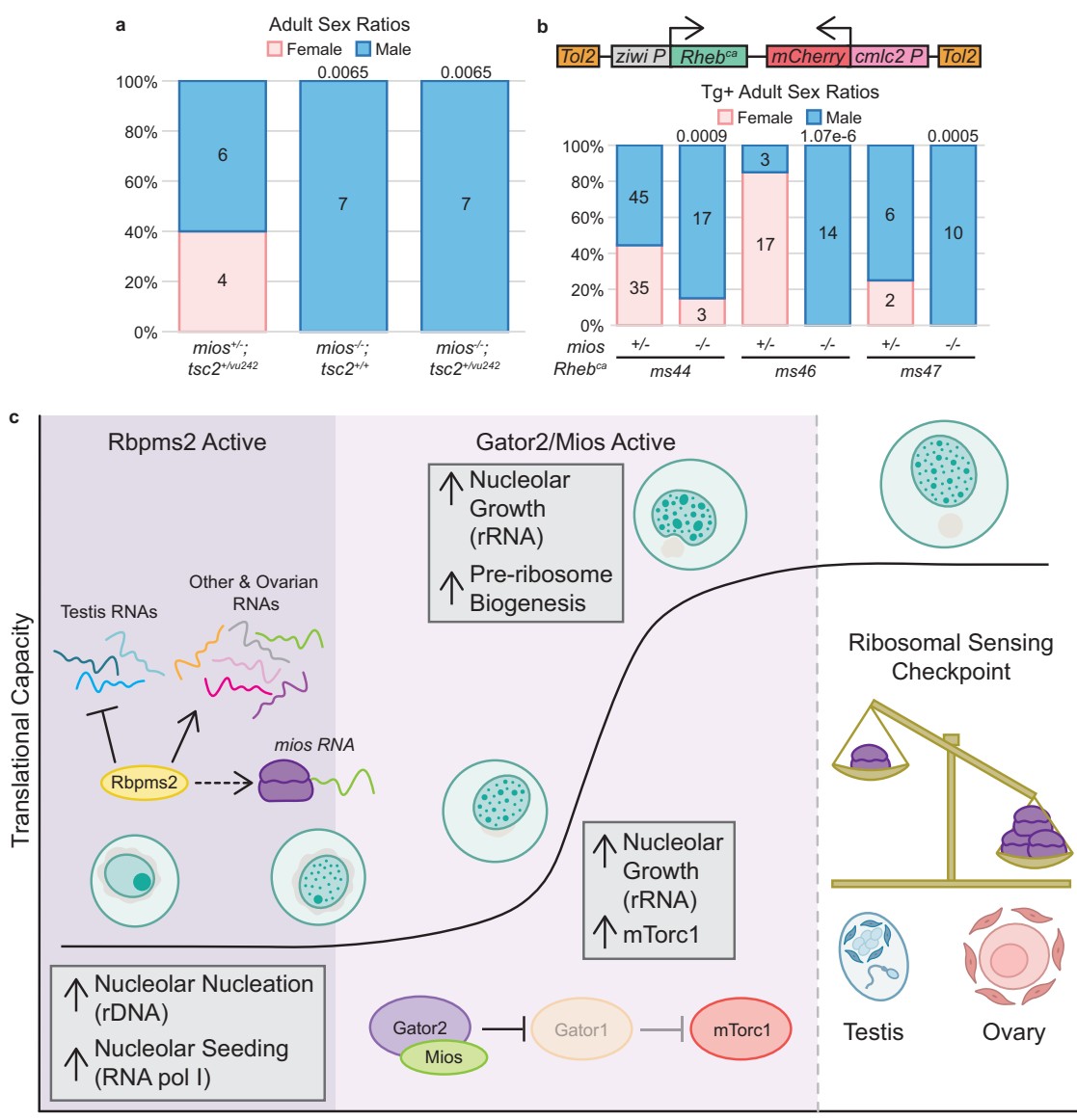

**Fig. 7 | mTorc1 activation in oogenesis uniquely requires Mios. a** Sex ratio graph for 60 dpf+ *mios; tsc2^vu242* double heterozygous, mutant wildtype, and mutant heterozygous fish. Pink represents female fish and blue indicates male fish. Number of fish screened are indicated for each group. Statistical significance was evaluated by performing a two-tailed $\chi^2$ test with Bonferroni correction against the *mios^+/-; tsc2^+vu242* control group. P = 0.0125. **b** *Rheb^ca* transgene: the rat *Rheb^ca* expression is driven by the germline *ziwi* promoter and the heart specific promoter *cmlc2* drives *mCherry* expression as a selectable marker. Sex ratios are presented for 60 dpf+ fish with the *ms44, ms46,* and *ms47 Rheb^ca* alleles. Pink represents female and blue represents males; number of fish screened are indicated for each group. Statistical significance was evaluated by performing the two-tailed $\chi^2$ test with Bonferroni correction against the +/- transgenic control group. P = 0.0125. Source data are provided as a Source Data file. **c** In mitotic and early meiotic cells, Rbpms2 functions to repress translation of *rboRNAs* related to testis fates and promotes

translation of *rboRNAs* related to mechanisms supporting oogenesis like ribosomal factors. Correspondingly, Rbpms2 functions upstream of nucleoli amplification and nucleolar localization of RNA pol I, which is required for rRNA transcription. In addition, Rbpms2 promotes translation of Mios. After sufficient nucleolar amplification, Gator2 and Mios activity increase mTorc1 signaling, and the nucleoli of differentiating oocytes expand and mature to support high levels of rRNA synthesis. As nucleoli grow, they develop an additional compartment that supports pre-ribosome biogenesis. Finally, as the cell progresses to diplotene, ribosomal abundance is measured. If sufficient ribosomes are present the cells continue through oogenesis. If a sufficient ribosome pool is not attained, then the gonocytes abort oogenesis and switch to spermatogenesis. We anticipate this pathway operates in oocytes independent of sex determination systems utilized as Mios is required for oocyte maturation in invertebrate and vertebrate organisms that use sex chromosome based or multigenic sex determination mechanisms.

the mechanisms that ensure sex-specific gamete differentiation, which in oocytes is orchestrated in part by mTorc1 signaling.

## Methods

### Fish strains
wildtype zebrafish embryos of the SAT strain were obtained from pairwise mating and were reared according to standard procedures[66]. Embryos were raised in 1X Embryo Medium at 28.5 °C and staged

accordingly[67]. *mios^ms20* mutant fish were generated using Crispr-Cas9 mutagenesis as detailed below[68]. The zebrafish *mios^sa22946* allele was obtained from the Sanger Institute's Zebrafish Mutation Project and acquired through Zebrafish International Resource Center (ZIRC)[43]. Complementation tests were performed by intercrossing carriers of *mios^sa22946* and *mios^ms20*. For epistasis analysis and to generate double mutants, *mios^sa22946* heterozygous or mutant fish were crossed to fish with the *spo11^vc73* or *tsc2^vu242* allele, generated in refs. 50 and 51,

respectively. All procedures and experimental protocols were performed in accordance with NIH guidelines and were approved by the Icahn School of Medicine at Mount Sinai Institutional (ISMMS) Animal Care and Use Committees (IACUC #2017-0114).

## Mutagenesis

The zebrafish *mios*[sa20] allele was generated by CRISPR-Cas9 mediated mutagenesis[68]. *mios* single-guide RNAs (sgRNA) targeting exon 2 were designed using the CHOPCHOP webtool[69]. Briefly, the gene-specific target and the constant oligonucleotides were annealed, and the overhangs were filled in using T4 DNA polymerase. In vitro transcription of the sgRNA was performed using the MEGAscript SP6 kit (Life technology, Ambion). 1 nL of 12.5 ng/μL of sgRNA and 1 nL of Cas9 protein (300 ng/μL)[70] along with phenol red (Sigma Aldrich) were co-injected in one-cell stage embryos. At 24 hpf, 8 injected embryos and 8 control uninjected siblings were assayed by PCR amplification and T7 endonuclease (New England Biolabs, M0302S) digest for mutation analysis[44]. Individuals with new gel banding patterns were sequenced to characterize induced mutations. Injected embryos were raised to adulthood and screened for germline mutations by extracting genomic DNA from their progeny and fin tissue and assaying as above. The presence of digested fragments compared to the wildtype allele indicated de novo mutations. The corresponding bands were extracted from the gel, cloned into a PCR4 TOPO vector (Invitrogen, K457502), and sequenced in both directions to determine the induced mutation. Fish harboring *mios*[ms20] mutations were outcrossed to SAT fish or crossed to *mios*[sa22946] for complementation tests. All mutations were verified by sequencing both genomic DNA (gDNA) and cDNA from mutant animals. Total RNA was extracted from individual gonads from heterozygote intercrosses or SAT wildtype, using TRIzol (Life Technologies, 15596). The SuperScript III/IV Reverse Transcription Kit (Life Technologies, 18080-051) was used to prepare cDNA, and Easy-A High Fidelity Taq polymerase (Agilent, 600400) was used for RT-PCR to amplify the *mios* coding region. Wildtype and mutant PCR fragments were TOPO cloned into pCR8/GW/TOPO (Invitrogen, K250020) and sequenced (Psomagen). Sequences were analyzed using SnapGene software. All oligos used are in Supplemental Table 1.

## Genotyping

gDNA was obtained either from embryos, dissected juvenile and adult trunks, or fin tissue. Samples were lysed in an alkaline lysis buffer (25 mM NaOH, 0.2 mM EDTA, pH12) and heated at 95 °C for 20 min. After cooling to 4 °C, neutralization buffer was added (20 mM Tris-HCl, 0.1 mM EDTA, pH 8.1)[71]. The genomic region around *rbpms2a*[ae30] was amplified for 35 cycles with an annealing temperature of 60 °C and the wildtype allele was digested with HaeIII for 1 h (New England Biolabs, R0108S)[2]. The *rbpms2b*[sa9329] surrounding genomic region was amplified for 35 cycles with an annealing temperature of 60 °C and the mutant allele was digested with MboII (New England Biolabs, R0148S)[2] or the wildtype allele was digested with HphI (New England Biolabs, R0158S) for 1 h. The region around the *mios*[sa22946] allele was amplified for 35 cycles using 60 °C for annealing and the mutant allele was identified by restriction enzyme digestion with DdeI for 1 h (New England BioLabs, R0175S). The genomic region around the *mios*[sa20] allele was amplified for 35 cycles using 57 °C for annealing and identified by restriction enzyme digestion of the wildtype allele with BsaWI for 1 h (New England BioLabs). The genomic region surrounding the *spo11*[uc73] allele was amplified for 35 cycles with an annealing temperature of 57 °C and products were resolved without digestion[50]. The surrounding *tsc2*[yu242] genomic region was amplified for 35 cycles with an annealing temperature of 57 °C and the wildtype allele was digested with HpyCH4IV for 1 h (New England Biolabs, R0169S). Undigested and digested products were resolved in a 3% Metaphor 1:1 (Lonza)/agarose gel (Invitrogen) gel. All genotyping primers are listed in Supplemental Table 1.

## Dissections and length measurements

Fish were anesthetized with a lethal dose of tricaine (MS-22). Fish were positioned laterally, and rostral-caudal body length was measured prior to dissection. Trunks and gonads were prepared by removal of the head and tail regions, exposure of the body cavity by opening the ventral body wall, and removal of the gut tissue while leaving the gonads intact. Alternatively, gonads were dissected from the trunk and reserved. Trunks and gonads were preserved by flash freezing in liquid nitrogen and storage at −80 °C or fixation as described below.

## Western blot

Individual gonads were dissected, lysed in 40 uL RIPA lysis buffer with protease inhibitor (Thermo Scientific, A32955) for 30 min at 4 °C, vortexed, and centrifuged at 4 °C. Lysates were removed and stored overnight at −80 °C. Sample lysates (18 uL) were then combined with equal volumes 2x SDS loading buffer and boiled for 5 min prior to loading. 20 μl per sample were loaded on a 4-12% SDS-PAGE gel and proteins were transferred to PVDF membranes. Membranes were blocked in 5% milk in 1x PBS for 1 h at room temperature. Anti-Mios antibody (Cell Signaling, 13557) was used at 1:250 and incubated overnight at 4 °C. Membranes were washed for 5 min 3 times in 1x TBS-Tween and then for 5 min twice in TBS. Rabbit-HRP secondary antibody (Cell Signaling, 7074) was diluted 1:5000. Following a 1 h incubation at room temperature, membranes were washed for 5 min 3 times 1x TBS-Tween and then for 5 min twice in 1x TBS. Proteins were detected with an ECL-Plus kit (Cytiva Lifescience, RPN2232). Chemiluminescence was imaged using a BioRad imager.

## Immunohistochemistry

For whole-mount immunofluorescence staining of gonads, tissues were fixed in 4% paraformaldehyde (PFA) overnight at 4 °C. The next day, the samples were washed in 1x PBS, dehydrated in 100% methanol (MeOH) overnight, and stored at −20 °C until use. Samples were gradually rehydrated with 1x PBS prior to acetone permeabilization at −20 °C for (1) 7 min and blocked in 1x PBSTw (0.1% Tween20 in 1x PBS) with 5% Normal Goat Serum and 2% dimethyl sulfoxide (DMSO) for 4 h minimum at 25 °C or (2) for 2 min and blocked in 1x PBSTw with 2% Normal Goat Serum and 2 mg/mL Bovine Serum Albumin for 4 h minimum at 25 °C. To stain for germ cells, samples were blocked (1 or 2) and the chicken anti-Vasa (Ddx4) antibody[50] was used at a 1:3000 dilution and incubated at 4 °C for 24 to 48 h. Nucleolar labeling and mTorc1 activity assessments were accomplished by blocking samples (1) and incubating with mouse anti-Fibrillarin (1:750 dilution; Novus Biologicals, NB300-269), rabbit anti-p-p70s6k (1:500 dilution; Cell Signaling, 9205), rabbit anti-Mios (1:500 dilution; Cell Signaling, 13557) at 4 °C for 24–48 h. RNA pol I and II visualizations were accomplished by blocking (2) and incubating samples with rabbit anti-p-UBTF (1:100 dilution; Invitrogen, PA5-105512) and mouse anti-RNA pol II (1:100 dilution; Santa Cruz, sc-56767) and incubated at 4 °C for 30 h minimum.

For all experiments, samples were washed with 1xPBSTw several times and incubated at 4 °C for 24–48 h with chicken Alexa Fluor 488, rabbit Alexa Fluor 568, and mouse Alexa Fluor 647 (Molecular Probes) secondary antibodies diluted 1:500 in (1) or 1x PBSTw (samples initially blocked in (2)). Samples were then washed in 1x PBSTw several times, dissected, and mounted in Vectashield with 4′,6-diamidino-2-phenylindole (DAPI) (Vector Laboratories), Prolong Diamond Antifade Mount with DAPI (Invitrogen), or incubated briefly (30 s to 15 min) in DAPI, washed with 1x PBSTw, and mounted in Prolong Diamond Antifade Mountant (Invitrogen). Images were acquired using a Zeiss Zoom dissecting scope equipped with Apotome.2, Zeiss Axio Observer inverted microscope equipped with Apotome.2 and a charged-coupled device camera, or a confocal microscope Zeiss 880 AiryScan2.0 (×40 or ×63 oil objective, 1 μM stack increments, 1024 × 1024

pixel format) at the Microscopy CoRE at ISMMS. Image processing was performed in Zen Blue (Zeiss), ImageJ/FIJI, and Adobe Illustrator.

## Fluorescence quantification

Fluorescence was quantified using ImageJ/Fiji by determining the area and mean intensity of cytoplasmic Ddx4 and the identical region occupied by the corresponding fluorophore. The average intensity for each was determined by dividing the mean intensity by the area and the average fluorescence intensity of p-Ps6k was then normalized to the average fluorescence intensity of Ddx4. Fluorescence was quantified from average intensity projections of 3, 1 μm slices and a minimum of 1 oocyte was quantified per fish. Fibrillarin puncta were quantified by generating maximum projections of 1 μm slices of whole oocytes. To define the edges of puncta, the brightness and contrast of the image was adjusted and then the Interactive Watershed plugin was used to better define the individual puncta with the following settings: Seed dynamics: 70, Intensity Threshold: 50, and Peak Flooding (%): 100. Splitting was allowed and the mask was exported to the image. The image was then thresholded and the particles were analyzed with a size cutoff of 5 pixels$^2$ to infinity to exclude artifacts. The area of the nucleus was determined by finding the center of the nucleus and measuring its area from a single slice. The puncta sizes were then normalized to the nuclear area of the given cell.

## RNA-IP

Wildtype ovaries were dissected from 90 dpf+ adult females from stable lines expressing oocyte-specific mApple tagged Rbpms2 protein (*Tg[buc:mApple-Rbpms2b-3'UTR]*; *n* = 2), or an oocyte-specific mApple only (*Tg[buc:mApple]*; *n* = 2) control line[2], snap frozen, and stored at -80 °C. We performed RNAIP adapted from previously published methods[23,72]. Briefly, ovaries were crosslinked in 250 μL YSS buffer (50 mM pH 8.0 Tris HCl, 50 mM NaCl, 0.1% NP40, 1 U/mL of RNAsin (Roche, 3335402001), protease inhibitor (Thermo Scientific, A32955), 0.5 mM DTT, 100 mM Sucrose) for 10 s in a UV crosslinker at 400mJx100cm$^2$ then homogenized. 750 μL YSS buffer was added to each homogenate and centrifuged. Pellets were resuspended in 1 mL YSS buffer. 250 μL of resuspended pellet was pre-cleared with 30 μL pre-washed protein G Dynabeads (Invitrogen, 10004D) for 1 h at 4 °C. 25 μL primary anti-RFP antibody (ChromoTek, 5f8) diluted 1:500 in YSS was added to each 250 μL pre-cleared lysate and incubated overnight at 4 °C with rocking. After incubation, 30 μL pre-washed Dynabeads were added to each sample and rocked at 4 °C for 2 h. At 4 °C, magnetic beads were washed five times with YSS buffer, resuspended in 100 μL Proteinase K lysis buffer (10 mM Tris HCl pH 8.0, 50 mM NaCl, 5 mM EDTA, 0.5% SDS) with 10 μg proteinase K (New England Biolabs, P8107S), vortexed briefly, and incubated at 50 °C for 1 h. Dynabeads were then removed, RNA was isolated using TRIzol (Life Technologies, 15596), and precipitated with 3 M sodium acetate. Precipitated RNA was stored at −80 °C until RNA library prep.

## RNAseq

For sequencing of RNAIP isolated RNAs, samples were processed (Qiagen, 74104) and cytoplasmic and mitochondrial rRNA-deleted stranded Total RNA libraries were prepared using Illumina Truseq RNA Library Prep kit (Illumina 20020589) from four individual adult ovaries (1 *buc:mApple-Rbpms2b-3'UTR* crosslinked, 1 *buc:mApple-Rbpms2b-3'UTR* uncrosslinked,1 *buc:mApple* crosslinked, and 1 *buc:mApple* uncrosslinked). Libraries were sequenced using an Illumina MiSeq, ~150 bp read length, paired ends, with ~ 1 million sequencing reads per sample. Sequencing data were aligned to the zebrafish genome GRCz10 using the Illumina website https://basespace.illumina.com. RNAs pulled down with mApple tagged Rbpms2 and mApple alone were compared from the cross-linked and uncrosslinked samples. Only RNAs that appeared in both the cross-linked and uncrosslinked

mApple-Rbpms2 samples, but not in mApple control samples were considered Rbpms2 target RNAs (*rboRNAs*).

For bulk RNA-sequencing, individual gonads were dissected at 21 dpf from *rbpms2* DMs (*n* = 4) and siblings (*rbpms2a^{ae30}*; *rbpms2b^{sa9329}* WM *n* = 4) and flash frozen in 100 μL of fresh RNA lysis buffer[23] containing 2-mercaptoethanol and stored at −80 °C prior to RNA purification. RNA was extracted and purified from individual tissues according to kit instructions (Agilent Absolutely RNA Nanoprep Kit, 400753), eluted in 13 μL RNA elution buffer, and stored at −20 °C prior to sequencing. RNA quality and quantity were assessed on an Agilent 2100 Bioanalyzer chip (RIN > 8). Libraries were generated and sequencing was performed by the NY Genome Technology Center with a low input SMART-Seq HT with Nxt HT kit (Clontech Laboratories, 634947) and SP100 cycle flow cell. All samples were sequenced on an Illumina SovaSeq 6000, paired end reads. Analysis of Fastq files and differential gene expression analyses were performed using BasePair pipelines. Further analyses were performed in R (v4.2.2). GO Term Biological Process Analysis was performed using clusterProfiler[73,74], GOfuncR[75], GOplot[76], genekitr[77], and rrvgo[78] packages and the volcano plot was generated using the EnhancedVolcano[79] package. Corresponding plots were generated with the ggplot2[80] package.

## Transgenesis

Generation of the *Tg[Tol2-ziwi:mTOR^{ca}; cmlc2:mCherry-Tol2]* and *Tg[Tol2-ziwi:Rheb^{ca}; cmlc2:mCherry-Tol2]* constructs was accomplished by using Gateway Recombination multiple-fragment cloning with the LR II+ Cloning Enzyme Mix (Invitrogen, 12538120). The *mTOR^{ca}* and *Rheb^{ca}* sequences (pME-mTOR L1460P[46]; pME-Rheb S16H[52]) were recombined with p5E-ziwiP[45] into the pBH-R4/R2 destination vector[23]. 1-cell embryos were injected with 1 nL of plasmid solution (100 ng of plasmid, 2.5 μL of Tol2 transposase RNA transcribed from pCS2FA-transposase, 0.25 μL of phenol red (Sigma Aldrich), and water (up to 5 μL)) and screened for mCherry heart expression at 1−3 dpf. Progeny of these fish were then screened for germline transmission of the given transgene by mCherry heart expression and stable lines were propagated. For accompanying analyses, progeny from at least 2 independent crosses were evaluated.

## Fertility assay

Female transgenic (*mios* wildtype *n* = 3, heterozygous *n* = 3, and mutant *n* = 3) siblings and non-transgenic (*mios* wildtype *n* = 3 and heterozygous *n* = 3) siblings for both the *ms49* and *ms64 mTOR^{ca}* alleles were crossed to transgenic male siblings, unrelated non-transgenic *mios* heterozygotes or mutants, or non-transgenic unrelated SAT fish weekly for 4−5 (*ms49* fish) or 3 (*ms64* fish) crosses total. All crosses were paired matings of individual males and females. All eggs were collected within two hours of initial mating and the number of total (degenerating + fertilized + unfertilized), degenerating, fertilized, and unfertilized eggs were counted. The fertility rate was calculated by dividing the number of fertilized eggs by the total number of eggs for a given cross and multiplying by 100.

## Electron microscopy

2 wildtype and 2 *rbpms2* DM 35 dpf fish were euthanized in tricaine (MS-22) and gonads were dissected and processed as previously described[2]. 4 *mios^{+/-}* and 4 *mios^{-/-}* 35 dpf fish were euthanized in tricaine (MS-22), dissected, and immediately fixed in Karnovsky's solution (2% glutaraldehyde and 2% paraformaldehyde in 0.1 M sodium cacodylate) for at least 24 h at 4 °C. The samples were then washed with cacodylate buffer, osmicated with 1% osmium tetroxide for 1 h, and en bloc stained with 2% uranyl acetate for 1 h. After a quick rinse in water and seven 10-min incremental ethanol dehydration steps (25%-100%), samples were infiltrated with a mixture of propylene oxide and an epoxy resin (Epon, Electron Microscopy Sciences). Samples were polymerized in pure resin in a vacuum oven at 60 °C for 72 h. Ultrathin sections were cut at

90 nm using a diamond knife (Diatome) on an ultramicrotome (Leica EM UC7) until the majority of cell nuclei were visible and mounted onto a formvar-supported slot grid (Electron Microscopy Sciences). Sections were imaged using a Hitachi H7500 TEM at 75 kV and 2048 × 2048 pixel, and 16-bit images of at least 10 oocytes per sample were taken using a CCD camera (AMT Imaging). Images were analyzed with FIJI/ImageJ and the Biodock Alligator model architecture[81].

### Statistics and reproducibility

Differential Gene Enrichment Analysis was done using the DESeq2 package with adjusted $P = 0.05$, adjusted P ($-\log_{10}$)=1.30, fold change of 2.00, and $\log_2$ fold change $= -1.00|1.00$. Sex ratio statistics were performed with a two-tailed $\chi^2$ test and a Bonferroni correction, with a confidence interval of 0.985, against stated groups in the associated figure legends. All fluorescence and fertility assay quantifications were done with a two-tailed, equal variance Student's $t$ test with a confidence interval of 0.95 between the stated groups in the associated figure legends. We calculated study power using group versus population parameters for both dichotomous endpoint as well as continuous mean endpoints with a type I error rate of 0.05 and power of 90%. No data were excluded from the analyses. All experiments were randomized, and Investigators were blinded to allocation during the experiments and outcome assessment where applicable.

### Reporting summary

Further information on research design is available in the Nature Portfolio Reporting Summary linked to this article.

## Data availability

All *rboRNA* analysis, bulk-RNA sequencing, nucleolar quantification, sex ratio, fluorescence, and fertility data are provided in the Source Data file. Source data are provided as a Source Data file. The RNAIP and bulk RNA raw sequencing data generated in this study have been deposited in the GEO database under the accession code GSE254850. Source data are provided with this paper.

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

## Acknowledgements

We thank the members of the Marlow and Rangan labs for helpful discussions and The Center for Comparative Medicine staff at ISMMS for fish care. Microscopy was performed at the Microscopy and Advanced Bioimaging CoRE at the Icahn School of Medicine at Mount Sinai. Bulk RNA sequencing was performed at the New York University Langone's Genome Technology Center (RRID: SCR_017929). We thank the Zebrafish International Resource Center for providing fish lines and Dr. Wenbiao Chen for providing the *mTOR* L1460P and *Rheb* S16H pME constructs. This work is supported by startup funds to F.L.M. and the National Institutes of Health (R01-GM089979), M.L.W. (1F31HD112158-01), S.R. (1F32HD097898-01A1), and O.K. (T32-GM007288, F30HD082903). S.R. was supported by a New York Stem Cell Foundation training grant (C32561GG). Work in the Draper lab was supported by R01 HD-081551 and NSF/IOS-1456737 to B.W.D.

## Author contributions

Conceptualization: M.L.W., S.R., O.K., and F.L.M. Methodology: M.L.W., S.R., Y.L., O.K., B.W.D., and F.L.M. Investigation: M.L.W., S.R., N.K., D.A., Y.L., O.K., and F.L.M. Visualization: M.L.W., S.R., N.K., D.A., Y.L., O.K., B.W.D., and F.L.M. Supervision: B.W.D. and F.L.M. Writing—original draft: M.L.W. and F.L.M. Writing—review and editing: M.L.W., S.R., N.K., D.A., Y.L., O.K., B.W.D., and F.L.M.

## Competing interests

The authors declare no competing interests.
