## [Peer Review File · Nature Communications]

Rbpms2 promotes female fate upstream of the nutrient sensing Gator2 complex component MiosREVIEWER COMMENTS

Reviewer #1 (Remarks to the Author):

The manuscript by Wilson et al., sheds light on the mechanism by which Rbmps2 acts as a necessary factor for oocyte development and subsequent female sex differentiation in the zebrafish. The authors show that Rbmps2 likely represses male differentiation factors while simultaneously promoting translation of ribosomal biogenesis factors and nucleolar amplification, which are critical for proper oocyte progression. Ultimately, the authors put forth an interesting model where Rbmps2 acts as a fate switch upstream of a nutrient sensing pathway, ultimately necessary for "pushing" a bipotential gonocyte towards an oocyte fate. In my opinion, this is a strong manuscript and the authors' claims are well supported. The experiments performed here are a logical progression from the previous work published by this group. The results flesh out the previous story and integrate nutrient signaling into the pathway. The findings will be a valuable contribution to the field of sex determination and germ cell biology.

Overall, the manuscript is well written. However, it was difficult to follow the logic in the section " Mios is required for oocyte differentiation independent of double strand break repair". Perhaps the authors could clarify this section by directly stating what the hypothesis is that they are trying to test. The manuscript would benefit from a clear 'working model' section in the discussion to bring everything together. The authors could consider incorporating something like the legend in figure 7C into the actual text.

Specific Comments:

- Define DMs at the beginning of the manuscript.
- Why is Pol II diminished in mios-/-? Could the authors please discuss this?
- It is difficult to see p-Ubft localized to nucleoli in Fig.4 h,I (much less to a distinct compartment)- a higher magnification would be helpful.

Reviewer #2 (Remarks to the Author):

- What are the noteworthy results?

- The identification of mRNAs that are bound by Rbpms2 proteins in zebrafish. This is important because Rbpms2 is required for the gonad to maintain an ovary fate and some of these transcripts may mediate those fate choices.
- Rbpms2 is required for nucleoli to proliferate during meiosis in oocytes.
- Rbpms2 acts via mTorc1 signaling.
- Rbpms2 is epistatic to mios.
- Without Rbpms2, testis factors are not repressed.
- Proper ribosome function is essential for ovary fate in zebrafish.

- Will the work be of significance to the field and related fields?

- yes.

How does it compare to the established literature?

Microscopy is very well done. The use of immunofluorescence is beautiful. Nice electron microscopy. Good use of various mutants and inducing new ones when they needed to. Good development of transgenic animals for specific purposes.

If the work is not original, please provide relevant references.
The work is original.

- Does the work support the conclusions and claims, or is additional evidence needed?

Some of the stated conclusions are logical inferences but are not the only possible explanations for the results. These are mentioned in the detailed reviewer comments and can be fixed in a revision.

- Are there any flaws in the data analysis, interpretation and conclusions?

Some conclusions fail to take into consideration alternative interpretations as noted in detailed comments.

- Do these prohibit publication or require revision?

The authors can easily revise the document to fix the perceived problems.

- Is the methodology sound? Does the work meet the expected standards in your field?

Yes. The methodology in general is sound. Two exceptions – first, the numbers of individuals checked for sex ratios in Figs. 5, and especially 6 and 7 are spectacularly low and no statistics are given, and second, no statistical analysis (unless I missed it) was performed to be sure that the identified mRNAs bound to Rbpms2 and not to the control are statistically robust. Nothing analogous to DESeq2 statistical output that one would see in an RNA-seq experiment although the methods are conceptually similar, seeing if the number of RNA molecules from a gene are statistically different in one situation vs. another.

- Is there enough detail provided in the methods for the work to be reproduced?

Yes.

Specific comments

Fig. 1A. The normal expression profiles of the two genes used for the transgenic analysis, rbpms2b and buc, need to be shown in Fig. 1B to compare to the other markers. Probably miso also because it plays such a pivotal role in the story.

Also, the expression pattern of rbpms2a and rbpms2b are essential with respect to cell type and time of expression, and their expression patterns in the scRNA-seq results.

P5. Define: DMs for non-zebrafish people. The introduction or start of the results should point out that zebrafish has two copies of this gene and the authors knocked out both.

P5. Was buc:mApple specifically expressed in oocytes according to microscopic observations?

P5. Tell the reader somewhere early at what specific stage in oogenesis rbpms2a and rbpms2b begin to be expressed. Especially relevant is when relative to the expression of the oocyte-specific gene buc, which provided the promoter for the transgene.

Likewise, readers need to know the expression pattern of rbpms2a and b in testis to understand the system. The human protein atlas shows that RBPMS2 is expressed in both spermatogonia and oocytes, but higher in oocytes. <https://www.proteinatlas.org/ENSG00000166831-RBPMS2/single+cell+type>.

Maybe it is also expressed in zebrafish spermatocytes. Public data might be available for this so new experiments would need to be done if the lab doesn't have the data already.

P5 The transgenic/immunoprecipitation method is a good one to identify interacting RNAs and the Fig. 1A does a good job of visually explaining the strategy.

P5. "The remaining 52 rboRNAs that did not map to the 40 dpf ovary dataset are likely transcripts expressed in later stage oocytes present in the fully mature adult ovary."

Yes, probably so, and that idea can be tested just by showing that they are or aren't previously characterized as late oocyte proteins from prior literature or public RNA-seq data.

Fig. 1 "(d) Volcano plot of differential gene expression in 21 dpf wildtype and rbpms2 DMs from bulk RNA sequencing."

This legend doesn't say what organs were taken, ovaries, testes, or whether these are whole animals.

P6 "Therefore, we conclude that 1) Rbpms2 likely functions to repress translation of rboRNAs

expressed in early gonocytes in ovaries and 2) promote translation of oocyte factors,'
Is this a conclusion or a hypothesis? From the data presented, it seems like these are logical hypotheses stemming from those data, but we can't actually conclude these two points from the presented data because translation was not specifically tested and other mechanisms are possible.

P6 'we analyzed GSEA enrichment plots'. Tell reader what these plots are, briefly.

P7 'Rbpms2 mediates a binary fate-switch by repressing testis factors and promoting nucleolar formation and oocyte development via mTor signaling'

The results do show that Rbpms2 promotes nucleolar formation, but the link to repressing testis factors is not really shown.

Fig. 2. Nice immunostaining images. It would be good for the legend to tell reader what the red nuage is.

Fig. 2 'wild-type (rbpms2^{aae30}; rbpms2^{bsa9329} HM; n=4) and rbpms2 DM'

Would be easier to understand if instead of HM it said double heterozygous and instead of DM it said double homozygous mutants.

pUbf1: Shouldn't the figure legend use p-Ubf1 like the text instead of p-Ubf1?

P7 'we reanalyzed transmission electron microscopy (TEM) images of rbpms2 DM oocytes'

Were these reanalyzed from a previous publication? If so, give citation here.

P7 'wild-type prophase I oocytes had greater than four nucleoli per cell at 35 dpf (Figure 2b, d) whereas most rbpms2 DM oocytes had fewer than 4 nucleoli (Figure 2c, d).'

Fig. 2a shows a great variation in nucleolus number for cells in different stages of prophase I in wildtypes. Which phases were present at 35dpf and which stage in meiosis were the cells in that were counted?

P8 'In rbpms2 DMs, while RNA pol I localized to the cytoplasm and nucleoli of mitotic germ cells, it was observed throughout oocyte nuclei and overlapped with RNA pol II nuclear localization (Figure 2f).'

Draw a conclusion here for that finding.

P8 'may do so by regulating translation of these factors to promote'

Conclusion is appropriate from the data, but I had to go back and reread the sentence to be sure of the antecedent to 'these'. It would help to say 'may do so by regulating translation of ribosome biogenesis factors to promote'.

P8. 'In wild-type ovaries, localization of phosphorylated p70-S6K (p-Ps6k), a kinase directly phosphorylated by the active form of mTorc1'

Do you think that readers should know that zebrafish has two copies of the relevant gene and that its official names are rps6kb1a and rps6kb1b?

Fig. 3 '(c) Table of mTorc1-related components with Rbpms2 binding sites in their 3' and/or 5' UTRs.'

Are these demonstrated Rbpms2 binding sites? Or are they sites predicted from some kind of consensus sequence?

P9 'wildtype, p-Ps6k was not detected in DM oocytes (Figure 3b). This observation suggests that dysregulated mTorc1 signaling in the absence of Rbpms2 contributes to impaired oogenesis.'

The data at this point show that Rbpms2 contributes to impaired mTorc1 signaling. It would take other experiments to show that it is the mTorc1 signaling that contributes to impaired oogenesis. It could be that it is one of the other things that's disrupted, like ribosome biogenesis, that contributes to impaired oogenesis and that the impaired mTorc1 signaling indeed happens but it's not the main reason oogenesis fails.

P9 'Missing oocyte (Mios), contains 4 Rbpms2 binding sites in its 3' UTR'

Again, are these proven binding sites? Or just predicted based on sequence?

P10 'Analysis of miosms20 and miossa22946 heterozygous and mutant progeny'

Unclear if 'heterozygous' here means miosms20/miossa22946 trans heterozygote or miosms20/mios+ and miossa22946/mios+.

P9 'Several in-frame deletions were recovered including miosms20, an 11bp deletion allele that leads to a frameshift'

Sentence says miosms20 is an in-frame deletion that is an 11bp deletion. But 11 is not divisible by 3 so how is this an in-frame deletion? Maybe it's a wording problem with the sentence.

P10. 'miosms20 and miossa22946 mutants and miosms20/sa22946 compound heterozygotes develop

functional testes and differentiate as males, exclusively'

Previously in the text, heterozygotes for a mutated allele were also called mutants. So here, text should specify whether 'miosms20 and miossa22946 mutants' means heterozygotes for each or homozygotes for each allele. The 'compound heterozygotes' phrase is clear.

P10. 'investigation at earlier timepoints,'

Earlier than what? From the writing, earlier than the stage indicated by 'Additionally, gonad morphological....' But the reader is not told what stage that is.

Fig. 4a Legend doesn't tell us that the asterisk mean stop for miossa/sa.

P10 'number of Fibrillar puncta but these puncta were much smaller'

Change to: number of Fibrillar puncta, but these were much smaller.

P10 'between wild-type and mios-/- oocytes (Figure 4h-i). In wildtype,'

Be consistent with 'wildtype', If you prefer to use 'wildtype' as one word when it is a noun, then you should use 'wildtype' as an adjective, e.g., 'this fish is a wildtype and this is a wildtype fish.' Only if you use 'wild type' as a noun should you use 'wild-type' as an adjective; e.g., 'this fish is a wild type and this is a wild-type fish'.

Fig. 5 Legend. It's unclear what 'specified cells' means in 'p-Ps6k and DAPI localization in specified cells'. Does it mean specified as germ cells vs. somatic gonadal cells? Specified as oocytes vs. spermatocytes? Or something else?

Fig. 5 Legend: 'Scale bar for (a,b) are 50 μ M'. Should be either scale bars are or scale bar is.

P10 'We generated two alleles, ms49 and ms64, of the mTORca transgene.'

Text should tell us what the sequence difference is between these two alleles.

P11. 'more total eggs than wildtype non-transgenic siblings'

This 'wildtype' usage as an adjective is correct if text consistently uses 'wildtype' as a noun to mean a genotype.

Fig. 5e. Numbers are quite small, six or seven or eight animals used to determine sex ratios. The conclusions are probably accurate, but numbers are quite small and no statistics are given to confirm that different genotypes are statistically different.

P11 'In fertility assays,'

Tell here briefly how these were done. Single pair matings? Wildtype AB or TU males? How many males per female? If different male genotypes were used, as the Methods section says, maybe some of the variation could be in the males.

Fig. 6 legend. 'HH, HW, and MH'

I couldn't find a definition of these abbreviations.

Fig. 5 legend. 'Number of fish screened are indicated'

Number is indicated. Or Numbers are indicated.

P13. 'Germline expression of Rhebca did not disrupt sex determination in wild-type genotypes nor prevent oocyte loss in mios-/- (Figure'

What does the text mean by 'wild-type genotypes'? If these animals are transgenic and expressing a constitutive allele of mouse Rheb, clearly they are not wildtype.

Fig. 7a, b. Numbers are extraordinarily small. I'm not sure that finding 3 males means that no females will be found. After all, there are lots of human families with three sons, but that doesn't mean that their next child won't be a daughter. No statistics are given to give confidence that the conclusions about sex ratios are valid.

Fig. 7c. Label the horizontal axis. Should the vertical axis have an arrowhead at the top?

P13. 'vertebrate specific RNAbp,'

Spell out RNA binding protein here for clarity.

P13 'Rbpms2 translationally represses testis-associated factors'

Remind reader here of the specific data that shows that translation of factors specific for testis development is repressed but translation of factors specific for ovary development are not repressed. This seems to be an important point of the paper and the data to make it need to be clearly stated and that a translation difference between the two gonad sexes is the main issue and not some other difference that results in different amounts of message or protein made.

The problem is that, if multitudes of ribosomes and high translation rates are required for oocytes to develop but spermatocytes can get by with fewer ribosomes and lower translation rates, that then

oogenesis will fail, and a failure of oogenesis by any of a variety of mechanisms, like blocking meiosis, also leads to testis development.

P13 'we demonstrate that the nutrient sensing arm of the mTorc1 pathway is uniquely required for oocyte progression and sustained oogenesis '

The result and conclusion are good, but could the effect be due to nutrient sensing difficulties in the soma – intestine, liver that makes vitellogenin, brain cells making gonadotrophin, etc. – rather than in the oocytes themselves? Is the effect cell autonomous to germ cells? Experiments can be designed to answer that question, but they are time consuming and difficult and publication shouldn't be held up for that, but the text should acknowledge that possibility.

P14. 'Additionally, RNA pol II, which we show is present in nuclei of wild-type zebrafish oocytes up to diplotene arrest, is also intact in rbpms2 DM oocytes suggesting that Rbpms2 likely regulates sexual differentiation through translational control.'

The result shows that Rbpms2 doesn't regulate sex differentiation by controlling the location or amount of RNA pol II. But that result doesn't show that Rbpms2 control is likely translational. It could also function on which genes are transcribed rather than just being in the nucleus at normal amounts, or on differential message stability, which also wouldn't necessarily change Rpol location and quantity. Then, the text can go on and rule out alternative explanations for the cited result.

P14. 'Specifically, several rboRNAs are associated with testis functions and their expression is limited to the early, undifferentiated cell types of the 40 dpf ovary.'

Do we know that those early, undifferentiated cell types at 40dpf are not already determined to be oocytes? I don't think the text told us when sex determination occurs in zebrafish. Is it before 40dpf? Also, the single cell transcriptomics for ovary should be contrasted to the single cell transcriptomics for testes of the same age to draw adequate conclusions.

P14 'This restricted expression is consistent with our hypothesis that Rbpms2 suppresses testis factors'

Yes, it is consistent, but other possibilities exist too. Factors other than Rbpms2 could be responsible for decreasing the number of testis associated RNAs after the undifferentiated cell type stage. And if there are no testis mRNAs, then they would not be there for Rbpms2 to bind whether or not Rbpms2 is responsible for them disappearing so only ovary associated mRNAs would be present for Rbpms2 to bind after ovary commitment. Some other ovary-promoting factor could be suppressing testis associated gene transcription or transcript stability early and so there are no testis associated mRNAs left for Rbpms2 to bind at later stages.

Do we know that Rbpms2 is expressed before the sex determination stage?

P15 'Notably, in zebrafish it has been shown that demethylation and amplification of an rDNA locus at the end of chromosome four (femrDNA) strongly correlates with female sex determination and differentiation⁵⁶.'

Could it be that the femrDNA and Rbpms2 act on sex determination in exactly the same way, by preventing the formation of the many ribosomes that are needed to make a mature oocyte? Can you rule out the model that 1. ribosomes are essential for making a mature oocyte; 2. without mature oocytes, some regulatory mechanism is disturbed that normally would prevent the ovary from transitioning to testis; 3. That there are several genes that are independently necessary, each in a different way, for making ribosomes functional, including both femrDNA and Rboms2.

P16. Did the zebrafish spo11 mutations block DSBs and meiosis? The text told us that homozygous mothers gave embryos that didn't do well but was it shown definitively that this was because DSBs didn't occur?

P16. 'inhibition of mTORC1 by conditional knockout'

Tell reader what the condition was in the conditional ko. Probably it was a cell-type specific, presumably oocyte specific ko, but in a couple of words the text would avoid making the reader go to the original paper just to find out.

P16. 'which in oocytes is orchestrated in part by mTorc1 signaling'

Do we know that that mTorc1 signaling is due to action in oocytes or is it non-autonomous due to effects in other cell types or in organs other than the gonad, like liver or gonadotrophin secreting brain cells?

P16. 'using Crispr-Cas9 mutagenesis as in as detailed below 62'

Fix wording.

P21. 'Sequencing data was aligned'

Either sequencing data were aligned or sequencing datum was aligned.

P21 'using the Illumina website'

Text could give here the URL.

P21. 'Only RNAs that appeared in both the cross-linked and uncrosslinked mApple-Rbpms2, but not in controls were considered Rbpms2 target RNAs.'

Good to have four replicates. Was there a minimum read count for concluding that a gene's transcripts were bound? Or was a single read sufficient to mean that a gene's transcripts were pulled down? Was there an adjusted p-value used to identify differentially pulled down transcripts between experimental and control?

P22 '(mios wildtype'

It would be less ambiguous to say homozygous mios wildtype, because it could have meant phenotypically wildtype, which would include heterozygotes. Likewise for homozygous mutants.

Reviewer #3 (Remarks to the Author):

Previous results implicate the RNA-binding protein Rbpms2 in ovary fate during zebrafish development. This work identifies RNA targets of Rbpms2 and, using a variety of approaches, supports a model whereby Rbpms2 promotes nucleolar amplification via TORC1, a step that supports oogenesis. In particular, the work examines the role of the GATOR2 component Mios in nucleolar development in oocytes, and its dependence on TOR to promote oogenesis independent of the TSC Rheb arm of TORC1. Nutrient availability had been implicated in oogenesis in other species, as well as in zebrafish sex determination. Impact of the work is high, for those interested in TOR signaling, germline development and the role of nutrients in oogenesis.

Several points listed below to clarify the results, strengthen the conclusions, and improve accessibility to a wider audience.

1. p.6-7: Refer to transcript abundance differences, rather than "stability". Elaborate on proposed feedback.
2. Throughout manuscript: Define all terms and abbreviations (e.g., DM, H, M, HH, HM, MW, MH)
3. Figure 2: define white arrows in legend.
4. Figure 3: add quantification to support conclusion of loss of Ps6k localization in rbpms2 DM; loss in mios mutant looks more convincing, but also needs quantification.
5. Figure 4g: add quantification to support differences in size of puncta. Presumably mios oocyte analysis was on the early arrested oocytes? Please clarify. Temper statement that mios is "required" for nucleologenesis since fibrillar staining shows nucleoli.
6. Figure 5: change or clarify labeling of x axes in legend (H, M, etc., change to genotype; state meaning of + and - for transgene).
7. Figure 7: correct the arrow from Gator1 to TORC as it should be negative.
8. Title of final results section is misleading. State more clearly result concerning amino acid sensing (GATOR-dependent) versus TSC/Rheb arm.
9. Supplemental Figure 6: stated phenotypes difficult to see; add zoom box.

REVIEWER COMMENTS

Reviewer #1 (Remarks to the Author):

The manuscript by Wilson et al., sheds light on the mechanism by which Rbpm2 acts as a necessary factor for oocyte development and subsequent female sex differentiation in the zebrafish. The authors show that Rbpm2 likely represses male differentiation factors while simultaneously promoting translation of ribosomal biogenesis factors and nucleolar amplification, which are critical for proper oocyte progression. Ultimately, the authors put forth an interesting model where Rbpm2 acts as a fate switch upstream of a nutrient sensing pathway, ultimately necessary for “pushing” a bipotential gonocyte towards an oocyte fate. In my opinion, this is a strong manuscript and the authors’ claims are well supported. The experiments performed here are a logical progression from the previous work published by this group. The results flesh out the previous story and integrate nutrient signaling into the pathway. The findings will be a valuable contribution to the field of sex determination and germ cell biology.

Overall, the manuscript is well written. However, it was difficult to follow the logic in the section “ Mios is required for oocyte differentiation independent of double strand break repair”. Perhaps the authors could clarify this section by directly stating what the hypothesis is that they are trying to test.

We have clarified the introduction in this section to make it clearer that we sought to determine if suppression of DSB formation could restore oogenesis in our *mios*^{-/-} fish as it does in *Drosophila mio*^{-/-}. We find that this is not the case and therefore conclude that this mechanism is not conserved in our vertebrate system.

The manuscript would benefit from a clear ‘working model’ section in the discussion to bring everything together. The authors could consider incorporating something like the legend in figure 7C into the actual text.

We thank the reviewer for this suggestion. We have added more detail to the discussion to more clearly state our overarching findings and hypotheses.

Specific Comments:

- Define DMs at the beginning of the manuscript.

We thank all reviewers for bringing this to our attention have defined *rbpm2* DMs.

- Why is Pol II diminished in *mios*^{-/-}? Could the authors please discuss this?

We noticed variation from sample to sample such that some oocytes had lower levels of RNA pol II in their nuclei as compared to other cells, but this was not different between genotypes. We have corrected the brightness/contrast of the image to accurately show the RNA pol II intensity.

- It is difficult to see p-Ubft localized to nucleoli in Fig.4 h,i (much less to a distinct compartment)– a higher magnification would be helpful.

We have modified the inset to show the individual p-Ubtf and RNA pol II channels and have outlined the RNA pol II excluded region to make the lack of the granular compartment in *mios* mutants clearer.

Reviewer #2 (Remarks to the Author):

- What are the noteworthy results?

- The identification of mRNAs that are bound by Rbpm2 proteins in zebrafish. This is important because Rbpm2 is required for the gonad to maintain an ovary fate and some of these transcripts may mediate those fate choices.
- Rbpm2 is required for nucleoli to proliferate during meiosis in oocytes.
- Rbpm2 acts via mTorc1 signaling.
- Rbpm2 is epistatic to *mios*.

- Without Rbpms2, testis factors are not repressed.
- Proper ribosome function is essential for ovary fate in zebrafish.

- Will the work be of significance to the field and related fields?
• yes.

How does it compare to the established literature?

Microscopy is very well done. The use of immunofluorescence is beautiful. Nice electron microscopy. Good use of various mutants and inducing new ones when they needed to. Good development of transgenic animals for specific purposes.

If the work is not original, please provide relevant references.
The work is original.

- Does the work support the conclusions and claims, or is additional evidence needed?
Some of the stated conclusions are logical inferences but are not the only possible explanations for the results. These are mentioned in the detailed reviewer comments and can be fixed in a revision.

- Are there any flaws in the data analysis, interpretation and conclusions?
Some conclusions fail to take into consideration alternative interpretations as noted in detailed comments.

- Do these prohibit publication or require revision?
The authors can easily revise the document to fix the perceived problems.

- Is the methodology sound? Does the work meet the expected standards in your field?
Yes. The methodology in general is sound. Two exceptions – first, the numbers of individuals checked for sex ratios in Figs. 5, and especially 6 and 7 are spectacularly low and no statistics are given, and second, no statistical analysis (unless I missed it) was performed to be sure that the identified mRNAs bound to Rbpms2 and not to the control are statistically robust. Nothing analogous to DESeq2 statistical output that one would see in an RNA-seq experiment although the methods are conceptually similar, seeing if the number of RNA molecules from a gene are statistically different in one situation vs. another.

We have addressed the sex ratio numbers for experiments in figures 5-7 and have performed corresponding statistical analyses. We observed 3 female *mios* mutants carrying the *ms44 Tg[zivi:Rheb^{ca}; cmlc2:mCherry]* transgenic insertion from a single cross out of 3 independent crosses that were evaluated for this allele. As female mutants were not observed among progeny from 2 other sibling crosses carrying the same transgenic insertion, we conclude that the suppression by Rheb^{ca} was incompletely penetrant within the *ms44* line. Furthermore, because suppression was not observed in 2 other independent alleles (representing different insertion events) we conclude that overexpression of Rheb^{ca} and therefore activation through this arm of the mTORc1 signaling pathway is not sufficient to restore oogenesis in *mios*^{-/-} fish.

We have added additional language to the RNAseq methods section to explain our *rboRNA* selection methodology. Specifically, we took an “all or none” approach - we only counted RNAs as bound if they were present in the mApple-Rbpms2 crosslinked and uncrosslinked samples and absent in mApple crosslinked and/or uncrosslinked controls. Therefore, enrichment statistics were not necessary for these RNAs. It is true that additional targets may be found if we were to perform a less stringent enrichment analysis.

- Is there enough detail provided in the methods for the work to be reproduced?
Yes.

Specific comments

Fig. 1A. The normal expression profiles of the two genes used for the transgenic analysis, *rpbms2b* and *buc*, need to be shown in Fig. 1B to compare to the other markers. Probably *mios* also because it plays such a pivotal role in the story.

Also, the expression pattern of *rpbms2a* and *rpbms2b* are essential with respect to cell type and time of expression, and their expression patterns in the scRNA-seq results.

We have revised the text to clarify the expression of Rbpms2 and Buckyball proteins. Previous work from our lab and others have demonstrated that both RNAs and proteins are present at the same time and in the early oocyte-specific structure, the Balbiani body. To facilitate comparisons of their RNA expression profiles, we have added the relevant RNA expression UMAPs to supplemental Figure 1, which demonstrates their highly similar expression profiles in the early ovary.

P5. Define: DMs for non-zebrafish people. The introduction or start of the results should point out that zebrafish has two copies of this gene and the authors knocked out both.

We thank the reviewer for noting this and have clarified our language to address the presence of the two genes and have defined “DM” in this context.

P5. Was buc:mApple specifically expressed in oocytes according to microscopic observations?

This promoter has been confirmed to express specifically in early-stage oocytes and drives mApple expression in oocytes specifically, as we have previously shown in Kaufman et al. *PLoS Genetics* 2014 (<https://doi.org/10.1371/journal.pgen.1007489>).

P5. Tell the reader somewhere early at what specific stage in oogenesis rbpms2a and rbpms2b begin to be expressed. Especially relevant is when relative to the expression of the oocyte-specific gene buc, which provided the promoter for the transgene.

We have clarified our language to make it clear that our previous work has shown that these proteins are both expressed at the same time and localize the Balbiani body of prophase I oocytes.

Likewise, readers need to know the expression pattern of rbpms2a and b in testis to understand the system. The human protein atlas shows that RBPMS2 is expressed in both spermatogonia and oocytes, but higher in oocytes. <https://www.proteinatlas.org/ENSG00000166831-RBPMS2/single+cell+type>. Maybe it is also expressed in zebrafish spermatocytes. Public data might be available for this so new experiments would need to be done if the lab doesn't have the data already.

We have previously shown in Romano et al. *Development* 2020 (<https://journals.biologists.com/dev/article/147/18/dev190942/225837/Loss-of-dmrt1-restores-zebrafish-female-fates-in>) that Rbpms2 is only detected in ovaries and we have clarified the text to make this clear.

P5 The transgenic/immunoprecipitation method is a good one to identify interacting RNAs and the Fig. 1A does a good job of visually explaining the strategy.

We thank the reviewer for this positive feedback 😊.

P5. “The remaining 52 rboRNAs that did not map to the 40 dpf ovary dataset are likely transcripts expressed in later stage oocytes present in the fully mature adult ovary.”

Yes, probably so, and that idea can be tested just by showing that they are or aren't previously characterized as late oocyte proteins from prior literature or public RNA-seq data.

Due to the cell size limitation of the FACS sorting performed on the zebrafish 40 dpf ovary, we could not analyze oocytes larger than ~70 µm. We have gone through the originally reported 52 RNAs that were not found in the 40 dpf dataset and have determined by searching updated gene names that only 7 of the 52 do not map to the 40 dpf dataset. We have searched for these 7 genes in other public oocyte datasets (<https://www.nature.com/articles/s41586-022-04918-4>) and have determined that all 7 are found in PGCs, oogonia, and oocytes of mouse and human fetal ovaries. Additionally, we find that 5 are present in an early zebrafish embryo dataset (<https://www.science.org/doi/10.1126/science.aar3131>), specifically at stages around when ZGA occurs, suggesting that these 4 RNAs are present in late oocyte stages. Only 2 could not be found in available databases or expression analyses. Figure 1 was modified based on these analyses. Specifically,

the panel of RNAs that were not enriched (Titled “None”) was removed and the figure was rearranged to accommodate this change.

Fig. 1 “(d) Volcano plot of differential gene expression in 21 dpf wildtype and *rbpms2* DMs from bulk RNA sequencing.”

This legend doesn’t say what organs were taken, ovaries, testes, or whether these are whole animals.

We have revised the figure legend to state that the bulk RNA sequencing was performed on bipotential gonads of wildtype and *rbpms2* DM 21 dpf fish.

P6 ‘Therefore, we conclude that 1) *Rbpms2* likely functions to repress translation of rbtRNAs expressed in early gonocytes in ovaries and 2) promote translation of oocyte factors,’

Is this a conclusion or a hypothesis? From the data presented, it seems like these are logical hypotheses stemming from those data, but we can’t actually conclude these two points from the presented data because translation was not specifically tested and other mechanisms are possible.

We have revised the text to clarify our language and to include the possibility of alternative mechanisms.

P6 ‘we analyzed GSEA enrichment plots’. Tell reader what these plots are, briefly.

We have added an explanatory statement to make this clear.

P7 ‘*Rbpms2* mediates a binary fate-switch by repressing testis factors and promoting nucleolar formation and oocyte development via mTor signaling’

The results do show that *Rbpms2* promotes nucleolar formation, but the link to repressing testis factors is not really shown.

We have expanded the discussion to include qualifying language to make clear that the proposed functions of *Rbpms2* as a translational repressor of testis RNAs and promoter of RNAs related to nucleoli formation and ribosome biogenesis are hypothesized functions given our results in this work and in the published literature.

Fig. 2. Nice immunostaining images. It would be good for the legend to tell reader what the red nuage is.

We have adjusted the images of the Mios protein localization to the cytoplasm of oocytes and have added magnified images of early oocyte stages. Additionally, we have added fluorescence quantification of Mios in wildtype and *rbpms2* DM oocytes to further support our claim that Mios protein is lacking in *rbpms2* DMs. In reference to the red nuage of figure 2, we are unclear what exactly is being noted here and have assumed this comment was regarding the Mios staining in figure 3.

Fig. 2 ‘wild-type (*rbpms2*^{aae30}; *rbpms2*^{bsa9329} HM; n=4) and *rbpms2* DM’

Would be easier to understand if instead of HM it said double heterozygous and instead of DM it said double homozygous mutants.

We have clarified the genotype information in each figure legend.

pUbf1: Shouldn’t the figure legend use p-Ubtf like the text instead of p-Ubf1?

We have corrected the discrepancy and have corrected the gene name to *ubtf* (Ubtf).

P7 ‘we reanalyzed transmission electron microscopy (TEM) images of *rbpms2* DM oocytes’

Were these reanalyzed from a previous publication? If so, give citation here.

We thank the reviewer for noting this and have updated the text with the corresponding reference.

P7 ‘wild-type prophase I oocytes had greater than four nucleoli per cell at 35 dpf (Figure 2b, d) whereas most *rbpms2* DM oocytes had fewer than 4 nucleoli (Figure 2c, d).’

Fig. 2a shows a great variation in nucleolus number for cells in different stages of prophase I in wildtypes. Which phases were present at 35dpf and which stage in meiosis were the cells in that were counted?

rbpms2 DM oocytes do not appear to reach diplotene but do progress through the early stages of prophase I as judged by Balbiani body (Bb) formation, so we analyzed early Bb stage wildtype and *rbpms2* DM cells that were in the stages of prophase I prior to arrest. We have highlighted a representative cell in the wildtype image to clarify this point.

P8 'In *rbpms2* DMs, while RNA pol I localized to the cytoplasm and nucleoli of mitotic germ cells, it was observed throughout oocyte nuclei and overlapped with RNA pol II nuclear localization (Figure 2f).'

Draw a conclusion here for that finding.

We have expanded upon this finding, noting that while RNA pol I localizes to 1-2 nucleoli in *rbpms2* DM oocytes, it also remains distributed throughout the oocyte nucleus (in contrast to wildtype) and suggests nucleolar seeding and consequently rRNA transcription is diminished.

P8 'may do so by regulating translation of these factors to promote'

Conclusion is appropriate from the data, but I had to go back and reread the sentence to be sure of the antecedent to 'these'. It would help to say 'may do so by regulating translation of ribosome biogenesis factors to promote'.

We thank the reviewer for their suggestion and have revised the sentence accordingly.

P8. 'In wild-type ovaries, localization of phosphorylated p70-S6K (p-Ps6k), a kinase directly phosphorylated by the active form of mTorc1'

Do you think that readers should know that zebrafish has two copies of the relevant gene and that its official names are *rps6kb1a* and *rps6kb1b*?

We have analyzed the amino sequences of the *Rps6kb1a* and *Rps6kb1b* zebrafish proteins against the human *Rps6k1b* and *Rps6k2b* proteins. Both zebrafish proteins are annotated as orthologs of the human *Rps6k1b*. Aligning the zebrafish proteins to both human proteins shows that the epitope target of the antibody is conserved in all cases, indicating that the antibody we used likely recognizes both zebrafish proteins. Further, single cell data indicates *rps6kb1a* is lowly expressed throughout the ovary (highest expression is in post-meiotic cells) and *rps6kb1b* is expressed throughout but is highest in GSCs/TA and oocyte cells. We have revised the text as recommended to clarify that there are two proteins in zebrafish.

Fig. 3 '(c) Table of mTorc1-related components with *Rbpms2* binding sites in their 3' and/or 5' UTRs.'

Are these demonstrated *Rbpms2* binding sites? Or are they sites predicted from some kind of consensus sequence?

We have clarified that these binding sites were predicted based on the published binding motif for the related family member *Rbpms*, as the exact binding motif of *Rbpms2* has not been precisely mapped. We have revised the text to include the citation to the work that characterized the *Rbpms* binding motif.

P9 'wildtype, p-Ps6k was not detected in DM oocytes (Figure 3b). This observation suggests that dysregulated mTorc1 signaling in the absence of *Rbpms2* contributes to impaired oogenesis.'

The data at this point show that *Rbpms2* contributes to impaired mTorc1 signaling. It would take other experiments to show that it is the mTorc1 signaling that contributes to impaired oogenesis. It could be that it is one of the other things that's disrupted, like ribosome biogenesis, that contributes to impaired oogenesis and that the impaired mTorc1 signaling indeed happens but it's not the main reason oogenesis fails.

We have revised the text to clarify that loss of *Rbpms2* leads to dysregulation of mTorc1 signaling, which may contribute to or be a consequence of defects in ribosome biogenesis and oogenesis.

P9 'Missing oocyte (Mios), contains 4 *Rbpms2* binding sites in its 3' UTR'

Again, are these proven binding sites? Or just predicted based on sequence?

These binding sites are predicted based on the validated Rbpms binding sites as described above.

P10 'Analysis of miosms20 and mioassa22946 heterozygous and mutant progeny'

Unclear if 'heterozygous' here means miosms20/mioassa22946 trans heterozygote or miosms20/mios+ and mioassa22946/mios+.

We have revised the text to clarify that the fish noted here are single heterozygotes or mutants for the indicated alleles.

P9 'Several in-frame deletions were recovered including miosms20, an 11bp deletion allele that leads to a frameshift'

Sentence says miosms20 is an in-frame deletion that is an 11bp deletion. But 11 is not divisible by 3 so how is this an in-frame deletion? Maybe it's a wording problem with the sentence.

Thank you for pointing this out. We did not word this well. We have revised the text to clarify that we found in-frame deletions as well as nonsense mutations including the *ms20* allele.

P10. 'miosms20 and mioassa22946 mutants and miosms20/sa22946 compound heterozygotes develop functional testes and differentiate as males, exclusively'

Previously in the text, heterozygotes for a mutated allele were also called mutants. So here, text should specify whether 'miosms20 and mioassa22946 mutants' means heterozygotes for each or homozygotes for each allele. The 'compound heterozygotes' phrase is clear.

We have revised the text and legends to indicate what "heterozygote" and "mutant" mean in the given contexts.

P10. 'investigation at earlier timepoints,'

Earlier than what? From the writing, earlier than the stage indicated by 'Additionally, gonad morphological....' But the reader is not told what stage that is.

We have revised the wording here to make clear that we also analyzed these fish prior to testis development.

Fig. 4a Legend doesn't tell us that the asterisk mean stop for mioassa/sa.

We have defined this in the revised figure legend.

P10 'number of Fibrillar in but these puncta were much smaller'

Change to: number of Fibrillar puncta, but these were much smaller.

We thank the reviewer for this suggestion and have made the correction.

P10 'between wild-type and mios-/- oocytes (Figure 4h-i). In wildtype,'

Be consistent with 'wildtype', If you prefer to use 'wildtype' as one word when it is a noun, then you should use 'wildtype' as an adjective, e.g., 'this fish is a wildtype and this is a wildtype fish.' Only if you use 'wild type' as a noun should you use 'wild-type' as an adjective; e.g., 'this fish is a wild type and this is a wild-type fish'.

We have revised the text to use "wildtype" for nouns and adjectives.

Fig. 5 Legend. It's unclear what 'specified cells' means in 'p-Ps6k and DAPI localization in specified cells'.

Does it mean specified as germ cells vs. somatic gonadal cells? Specified as oocytes vs. spermatocytes? Or something else?

We have revised the wording here to "indicated cells" rather than specified.

Fig. 5 Legend: 'Scale bar for (a,b) are 50 μ M'. Should be either scale bars are or scale bar is.

We have made this correction.

P10 'We generated two alleles, ms49 and ms64, of the mTORca transgene.'
Text should tell us what the sequence difference is between these two alleles.

We have revised the sentence to indicate that these alleles are the same transgene and that the two alleles only differ by independent insertion events in the genome.

P11. 'more total eggs than wildtype non-transgenic siblings'
This 'wildtype' usage as an adjective is correct if text consistently uses 'wildtype' as a noun to mean a genotype.

We have made this correction.

Fig. 5e. Numbers are quite small, six or seven or eight animals used to determine sex ratios. The conclusions are probably accurate, but numbers are quite small and no statistics are given to confirm that different genotypes are statistically different.

We have updated all sex ratio graphs with corresponding statistics and increased numbers as indicated above.

P11 'In fertility assays,'
Tell here briefly how these were done. Single pair matings? Wildtype AB or TU males? How many males per female? If different male genotypes were used, as the Methods section says, maybe some of the variation could be in the males.

We have revised the methods section to indicate that these crosses were single paired matings, and to indicate that the wildtype fish were SATs. The females analyzed were not crossed to the same male every instance. Although we cannot exclude that the males did contribute to the results we see, but we did not see significant differences among females (of the same mutant and transgenic genotypes) from mating to mating, but differences were seen between the *mios* mutant Tg+ females compared to nonmutant Tg+ and Tg- female siblings. Thus, we conclude that the effects seen are due to the transgene in the context of lack of Mios. Additionally, we did compare the Tg+ and Tg- wildtype and heterozygous siblings to each other and found that these siblings, as expected, were not statistically different from each other, suggesting that the male-to-male variation didn't contribute significantly to the results seen.
We have also updated the text with a brief description of how the fertility assay was conducted.

Fig. 6 legend. 'HH, HW, and MH'
I couldn't find a definition of these abbreviations.

We have included these definitions in the revised manuscript.

Fig. 5 legend. 'Number of fish screened are indicated'
Number is indicated. Or Numbers are indicated.

We have made this correction.

P13. 'Germline expression of Rhebca did not disrupt sex determination in wild-type genotypes nor prevent oocyte loss in *mios*^{-/-} (Figure'
What does the text mean by 'wild-type genotypes'? If these animals are transgenic and expressing a constitutive allele of mouse Rheb, clearly they are not wildtype.

We have revised the language here.

Fig. 7a, b. Numbers are extraordinarily small. I'm not sure that finding 3 males means that no females will be found. After all, there are lots of human families with three sons, but that doesn't mean that their next child won't be a daughter. No statistics are given to give confidence that the conclusions about sex ratios are valid.

We have increased the numbers of animals examined for the *ms44* and *ms47* alleles. We observed 3 female *mios* mutants carrying the *ms44 Tg[ziwi:Rheb^{ca}; cmlc2:mCherry]* transgenic insertion from a single cross out of 3 independent crosses that were evaluated for this allele. As female mutants were not observed among progeny from 2 other sibling crosses carrying the same transgenic insertion, we conclude that the suppression by *Rheb^{ca}* was incompletely penetrant within the *ms44* line. The data for these crosses are included as Supp. Figure 8 in the revised manuscript. Furthermore, because suppression was not observed in 2 other independent alleles (representing different insertion events) we conclude that overexpression of *Rheb^{ca}* and therefore activation through this arm of the mTorc1 signaling pathway is not sufficient to restore oogenesis in *mios*^{-/-} fish.

Fig. 7c. Label the horizontal axis. Should the vertical axis have an arrowhead at the top?

We have corrected this in the revised manuscript.

P13. 'vertebrate specific RNAbp,'
Spell out RNA binding protein here for clarity.

We have revised this as recommended.

P13 'Rbpms2 translationally represses testis-associated factors'

Remind reader here of the specific data that shows that translation of factors specific for testis development is repressed but translation of factors specific for ovary development are not repressed. This seems to be an important point of the paper and the data to make it need to be clearly stated and that a translation difference between the two gonad sexes is the main issue and not some other difference that results in different amounts of message or protein made. The problem is that, if multitudes of ribosomes and high translation rates are required for oocytes to develop but spermatocytes can get by with fewer ribosomes and lower translation rates, that then oogenesis will fail, and a failure of oogenesis by any of a variety of mechanisms, like blocking meiosis, also leads to testis development.

We have tried to emphasize these points in the appropriate sections of the revised text and hope it is more clear now.

P13 'we demonstrate that the nutrient sensing arm of the mTorc1 pathway is uniquely required for oocyte progression and sustained oogenesis' 'The result and conclusion are good, but could the effect be due to nutrient sensing difficulties in the soma – intestine, liver that makes vitellogenin, brain cells making gonadotrophs, etc. – rather than in the oocytes themselves? Is the effect cell autonomous to germ cells? Experiments can be designed to answer that question, but they are time consuming and difficult and publication shouldn't be held up for that, but the text should acknowledge that possibility.'

We thank the reviewer for pointing out this alternative. Because we are able to restore oogenesis by expressing mTOR^{ca} only in germ cells under the germline-specific promoter, *ziwi*, this indicates that restoring activity in the germ cells, including oocytes, is sufficient. If required in the somatic cells this activity is likely regulated independent of *Mios*.

P14. 'Additionally, RNA pol II, which we show is present in nuclei of wild-type zebrafish oocytes up to diplotene arrest, is also intact in *rbpms2* DM oocytes suggesting that *Rbpms2* likely regulates sexual differentiation through translational control.'

The result shows that *Rbpms2* doesn't regulate sex differentiation by controlling the location or amount of RNA pol II. But that result doesn't show that *Rbms2* control is likely translational. It could also function on which genes are transcribed rather than just being in the nucleus at normal amounts, or on differential message stability, which also wouldn't necessarily change Rpol location and quantity. Then, the text can go on and rule out alternative explanations for the cited result.

Our conclusion that these oocytes are not failing due to transcriptional dysregulation is based on the RNAseq data showing that neither 1) RNA levels of *rboRNAs* nor 2) recruitment of the RNA pol II machinery to the

nucleus change between wildtype and *rbpms2* DM fish. Therefore, we conclude that Rbpms2 likely regulates translation of *rboRNAs* since RNA abundance appears unperturbed. For example, because we did not see significant changes in *mios* abundance between wildtype and *rbpms2* DMs, but we did see that *rbpms2* DM oocytes have significantly less Mios protein, this suggests, and we hypothesize that Rbpms2 promotes its translation.

P14. 'Specifically, several *rboRNAs* are associated with testis functions and their expression is limited to the early, undifferentiated cell types of the 40 dpf ovary.'

Do we know that those early, undifferentiated cell types at 40dpf are not already determined to be oocytes? I don't think the text told us when sex determination occurs in zebrafish. Is it before 40dpf? Also, the single cell transcriptomics for ovary should be contrasted to the single cell transcriptomics for testes of the same age to draw adequate conclusions.

Because these cells underwent single cell sequencing, we can say by the RNAs present that these cells are still bipotential and have the capacity to differentiate into oocytes or spermatocytes. Because Rbpms2 protein is detected in the ovary, but not the testis we assume that *rbtRNAs* in the testis are available for translation in the absence of Rbpms2. We have revised the text to clarify the significance of the observation that *rbtRNAs* are restricted to early, undifferentiated cells of the ovary. We have also revised the text to better explain sex determination and differentiation in zebrafish as well as the plasticity of the zebrafish ovary.

P14 'This restricted expression is consistent with our hypothesis that Rbpms2 suppresses testis factors' Yes, it is consistent, but other possibilities exist too. Factors other than Rbpms2 could be responsible for decreasing the number of testis associated RNAs after the undifferentiated cell type stage.

We thank the reviewer for pointing this out. We have revised the text to include the possibility that Rbpms2 may directly or indirectly influence *rboRNAs* and aspects of oogenesis through yet to be determined binding partners.

And if there are no testis mRNAs, then they would not be there for Rbpms2 to bind whether or not Rbpms2 is responsible for them disappearing so only ovary associated mRNAs would be present for Rbpms2 to bind after ovary commitment.

Because we identified the *rbtRNAs* in wildtype oocytes, this does indicate that in a wildtype context the testis RNAs are present and available for Rbpms2 to bind and regulate, even after ovary commitment. Continuous RNA expression and maintenance in a repressed state is common in oocytes and other several cell types, and we think this state is likely key to plasticity of the ovary. Accordingly, maintaining but repressing these testis RNAs in early germ cells of the ovary, as we think Rbpms2 does, would be important for sustained oocyte development and maintenance of the ovary.

Some other ovary-promoting factor could be suppressing testis associated gene transcription or transcript stability early and so there are no testis associated mRNAs left for Rbpms2 to bind at later stages.

We agree that other ovary-promoting factors likely suppress transcription of testis associated genes in differentiated oocytes at later stages. However, because *rbtRNAs* were found bound to Rbpms2 in adult ovary, and because their abundance did not change in early oocytes of *rbpms2* DMs, and because loss of *rbpms2* results in testis development, we think Rbpms2 translationally represses these testis RNAs in early oocytes. Because *rbpms2* DM oocytes do not make it past prophase I, we cannot make any claims about the role of Rbpms2 and its potential interacting partners in later oocyte stages. In the revised discussion, we have added the possibility that loss of Rbpms2 may have direct and indirect effects on oogenesis.

Do we know that Rbpms2 is expressed before the sex determination stage?

Our bulk sequence analysis indicates that *rbpms2* transcripts are present in 21 day and 28 day gonads, stages prior to sex determination. Single cell analysis indicates that *rbpms2* transcripts are detected in some mitotic cells but are most highly expressed in early meiotic cells and differentiating oocytes (we have added UMAP plots of *rbpms2a* and *rbpms2b* expression in the 40 dpf). However, Rbpms2 protein is not detected in mitotic

germ cells, but is present in meiotic cells and differentiating oocytes, indicating that *rbpms2* expression and activity increase as the cells differentiate.

P15 'Notably, in zebrafish it has been shown that demethylation and amplification of an rDNA locus at the end of chromosome four (*femrDNA*) strongly correlates with female sex determination and differentiation⁵⁶.' Could it be that the *femrDNA* and *Rbpms2* act on sex determination in exactly the same way, by preventing the formation of the many ribosomes that are needed to make a mature oocyte?

We agree that both *femrDNA* and *Rbpms2* could both converge on regulation of ribosomes. Because rDNAs are required for ribosome nucleation and because *femrDNA* is unmethylated in ovary, *femrDNA* would promote increased nucleation of nucleoli in oocytes, a prerequisite for Poll recruitment. Because we see reduced Poll in *rbpms2* DMs, it is tempting to speculate that *Rbpms2* might indirectly regulate *femrDNA* accessibility and are keen to test this possibility. Given the importance of rDNA for nucleation, we hypothesize that *Rbpms2* would be positively influencing expression from the *femrDNA* locus to promote ribosome biogenesis through direct or indirect mechanisms or act downstream of *femrDNA* modification. However, we feel that testing these notions, although exciting, is beyond the scope of this work.

Can you rule out the model that 1. ribosomes are essential for making a mature oocyte;

We agree that ribosomes are essential for making mature oocytes.

2. without mature oocytes, some regulatory mechanism is disturbed that normally would prevent the ovary from transitioning to testis;

We agree that mature oocytes are important; however, in the absence of *Rbpms2* oogenesis is perturbed at stages before mature oocytes are present in wild-type; thus, we think it plays a central role in differentiation.

3. That there are several genes that are independently necessary, each in a different way, for making ribosomes functional, including both *femrDNA* and *Rboms2*.

Without additional experiments, we cannot exclude that regulation of *femrDNA* is independent of *Rbpms2*. Further, although likely important, *femrDNA* modification has yet to be shown to be essential in functional assays. As mentioned above, our findings that ribosome biogenesis may be impaired by reduced RNA pol I recruitment, we hypothesize that promoting *femrDNA* locus accessibility (and thus nucleolar nucleation) could require *Rbpms2* through direct or indirect mechanisms.

P16. Did the zebrafish *spo11* mutations block DSBs and meiosis? The text told us that homozygous mothers gave embryos that didn't do well but was it shown definitively that this was because DSBs didn't occur?

The allele we utilized has been previously published and has shown to prevent DSBs in zebrafish testes, and to cause aneuploidy in oocytes. According, it is likely that DSBs are correspondingly blocked in zebrafish oocytes.

P16. 'inhibition of mTORC1 by conditional knockout'

Tell reader what was the condition was in the conditional ko. Probably it was a cell-type specific, presumably oocyte specific ko, but in a couple of words the text would avoid making the reader go to the original paper just to find out.

We have revised the text to clarify that this conditional knockout was in the mouse testis.

P16. 'which in oocytes is orchestrated in part by mTorc1 signaling'

Do we know that that mTorc1 signaling is due to action in oocytes or is it non-autonomous due to effects in other cell types or in organs other than the gonad, like liver or gonadotrophin secreting brain cells?

As mentioned above, because mTOR^{ca} expression in germ cells alone (expressed using the germ cell specific *ziwi* promoter) was sufficient to restore oogenesis and support female sex differentiation, we hypothesize that this activity is essential in oocytes.

P16. 'using Crispr-Cas9 mutagenesis as in as detailed below 62'
Fix wording.

We have made this correction.

P21. 'Sequencing data was aligned'
Either sequencing data were aligned or sequencing datum was aligned.

We have corrected this in the revised manuscript.

P21 'using the Illumina website'
Text could give here the URL.

The URL, <https://basespace.illumina.com>, has been added as recommended.

P21. 'Only RNAs that appeared in both the cross-linked and uncrosslinked mApple-Rbpms2, but not in controls were considered Rbpms2 target RNAs.'

Good to have four replicates. Was there a minimum read count for concluding that a gene's transcripts were bound? Or was a single read sufficient to mean that a gene's transcripts were pulled down? Was there an adjusted p-value used to identify differentially pulled down transcripts between experimental and control?

We have added additional language to the RNAseq methods section to explain our *rboRNA* selection methodology. Specifically, we took an "all or none" approach - we only counted RNAs as bound if they were present in the mApple-Rbpms2 crosslinked and uncrosslinked samples and absent in mApple crosslinked and/or uncrosslinked controls.

P22 '(mios wildtype'

It would be less ambiguous to say homozygous mios wildtype, because it could have meant phenotypically wildtype, which would include heterozygotes. Likewise for homozygous mutants.

We have stated all of the genotypes used because we evaluated effects of the transgene on heterozygous and mutant fish compared to their homozygous wildtype siblings.

Reviewer #3 (Remarks to the Author):

Previous results implicate the RNA-binding protein Rbpms2 in ovary fate during zebrafish development. This work identifies RNA targets of Rbpms2 and, using a variety of approaches, supports a model whereby Rbpms2 promotes nucleolar amplification via TORC1, a step that supports oogenesis. In particular, the work examines the role of the GATOR2 component Mios in nucleolar development in oocytes, and its dependence on TOR to promote oogenesis independent of the TSC Rheb arm of TORC1. Nutrient availability had been implicated in oogenesis in other species, as well as in zebrafish sex determination. Impact of the work is high, for those interested in TOR signaling, germline development and the role of nutrients in oogenesis.

Several points listed below to clarify the results, strengthen the conclusions, and improve accessibility to a wider audience.

1. p.6-7: Refer to transcript abundance differences, rather than "stability". Elaborate on proposed feedback.

We thank the reviewer for the suggestion and have revised the text accordingly.

2. Throughout manuscript: Define all terms and abbreviations (e.g., DM, H, M, HH, HM, MW, MH)

We thank the reviewer for noting this and have defined abbreviations in the revised manuscript.

3. Figure 2: define white arrows in legend.

We have defined this in the revised manuscript.

4. Figure 3: add quantification to support conclusion of loss of Ps6k localization in *rbpms2* DM; loss in *mios* mutant looks more convincing, but also needs quantification.

We thank the reviewer for this suggestion and have quantified the p-Ps6k and Mios in *rbpms2* DMs and added plots to Figure 3.

In the revised text, figures 3 and 5 were modified from an overview image of p-Ps6k in wildtype and *rbpms2* DMs or *mios*^{-/-} cells to magnified images of the nuclear localization in mitotic/early meiotic cells and oocytes to make the findings clearer. Overview images corresponding to those used for the magnified views are now in Supplemental Figure 2.

In figure 3, the overall brightness of the channels in the panels of the Mios staining was adjusted, and the adjacent panels that showed only Mios and DAPI were replaced with insets to make the phenotypes clearer to viewers. Further, quantification of Mios protein in wildtype and *rbpms2* DMs was added for consistency with the other quantifications performed (this data is in Figure 3 and Supp. Figure 6). In addition, the RNA names for Ps6k to *rps6kb1a* and *rps6kb1b* were corrected (they were originally written as *rpsk6kb1a* and *rpsk6kb1b*).

5. Figure 4g: add quantification to support differences in size of puncta. Presumably *mios* oocyte analysis was on the early arrested oocytes? Please clarify. Temper statement that *mios* is “required” for nucleogenesis since fibrillar staining shows nucleoli.

We have added the quantification of the fibrillar puncta to Figure 4 and have added quantification of the nuclear sizes of the cells used for the analysis to supplemental figure 6. Further, we have revised the language from “required for nucleogenesis” to “nucleolar maturation” based on the data.

In Figure 4, the overall brightness of the Fibrillar staining was increased to make the phenotypes clearer to viewers.

In addition, we corrected the x-axis labels for plots Supp. Figure 4 g-h and the y-axis of Supp. Figure 6d due to scaling errors. These changes did not change the representation of the data.

6. Figure 5: change or clarify labeling of x axes in legend (H, M, etc., change to genotype; state meaning of + and – for transgene).

We have revised the use of H, M, etc. to the +/+, +/-, etc. convention. We have also revised the figure legend to clarify which fish are transgenic and non-transgenic.

7. Figure 7: correct the arrow from Gator1 to TORC as it should be negative.

Thank you. This has been corrected in the revised manuscript.

8. Title of final results section is misleading. State more clearly result concerning amino acid sensing (GATOR-dependent) versus TSC/Rheb arm.

We have edited this title and revised the concluding statement regarding the unique Gator2-Mios mediated activation of mTorc1 signaling in zebrafish oocytes.

9. Supplemental Figure 6: stated phenotypes difficult to see; add zoom box.

We thank the reviewer for this feedback and have made the recommended changes.

REVIEWERS' COMMENTS

Reviewer #1 (Remarks to the Author):

The authors have responded carefully to reviewers' concerns. The work will be a significant contribution to the field.

In the abstract, it is a bit of a stretch to state that the mammalian gonad is initially an ovary. It would be more accurate to state that it has a minor ovarian bias.

Reviewer #2 (Remarks to the Author):

The authors have responded to all suggestions.

Reviewer #3 (Remarks to the Author):

The revisions have satisfactorily addressed my previous comments.